# Amoxicillin-resistant *Streptococcus pneumoniae* can be resensitized by targeting the mevalonate pathway as indicated by sCRilecs-seq

Liselot Dewachter[1], Julien Dénéréaz[1], Xue Liu[1,2], Vincent de Bakker[1], Charlotte Costa[3], Mara Baldry[3], Jean-Claude Sirard[3], Jan-Willem Veening[1]*

[1]Department of Fundamental Microbiology, Faculty of Biology and Medicine, University of Lausanne, Biophore Building, Lausanne, Switzerland; [2]Guangdong Key Laboratory for Genome Stability and Human Disease Prevention, Department of Pharmacology, International Cancer Center, Shenzhen University Health Science Center, Shenzhen, China; [3]Univ. Lille, CNRS, Inserm, CHU Lille, Institut Pasteur Lille, U1019 - UMR 9017 - CIIL - Center for Infection and Immunity of Lille, Lille, France

*For correspondence:
Jan-Willem.Veening@unil.ch

Competing interest: The authors declare that no competing interests exist.

**Abstract** Antibiotic resistance in the important opportunistic human pathogen *Streptococcus pneumoniae* is on the rise. This is particularly problematic in the case of the β-lactam antibiotic amoxicillin, which is the first-line therapy. It is therefore crucial to uncover targets that would kill or resensitize amoxicillin-resistant pneumococci. To do so, we developed a genome-wide, single-cell based, gene silencing screen using CRISPR interference called sCRilecs-seq (subsets of CRISPR interference libraries extracted by fluorescence activated cell sorting coupled to next generation sequencing). Since amoxicillin affects growth and division, sCRilecs-seq was used to identify targets that are responsible for maintaining proper cell size. Our screen revealed that downregulation of the mevalonate pathway leads to extensive cell elongation. Further investigation into this phenotype indicates that it is caused by a reduced availability of cell wall precursors at the site of cell wall synthesis due to a limitation in the production of undecaprenyl phosphate (Und-P), the lipid carrier that is responsible for transporting these precursors across the cell membrane. The data suggest that, whereas peptidoglycan synthesis continues even with reduced Und-P levels, cell constriction is specifically halted. We successfully exploited this knowledge to create a combination treatment strategy where the FDA-approved drug clomiphene, an inhibitor of Und-P synthesis, is paired up with amoxicillin. Our results show that clomiphene potentiates the antimicrobial activity of amoxicillin and that combination therapy resensitizes amoxicillin-resistant *S. pneumoniae*. These findings could provide a starting point to develop a solution for the increasing amount of hard-to-treat amoxicillin-resistant pneumococcal infections.

## Editor's evaluation

This paper will have a high impact on screening strategies, on factors that determine cell shape and peptidoglycan synthesis in pneumococcus and other bacteria, and on antibiotic potentiation to overcome resistance.

**eLife digest** *Streptococcus pneumoniae* is a bacterium that can cause pneumonia, meningitis and other life-threatening illnesses in humans. Currently, many *S. pneumoniae* infections are treated with the antibiotic amoxicillin, which kills the bacteria by weakening a structure known as the cell wall that surrounds each bacterium. However, more and more *S. pneumoniae* cells are becoming resistant to amoxicillin, making it harder to treat such infections.

We need new ways to effectively treat *S. pneumoniae* infections in humans. One potential strategy would be to combine amoxicillin with another drug that boosts the activity of amoxicillin so that it is able to kill the resistant bacteria. Two drugs that both target the same process in cells are more likely to boost each other's activity. Therefore, Dewachter et al. decided to search for another drug that also weakens the cell wall of *S. pneumoniae*.

The team first developed a new screening approach called sCRilecs-seq to silence individual genes in single *S. pneumoniae* cells. By looking at many cells that each had a different gene that was no longer active, the team were able to identify several genes that when silenced resulted in the cells becoming longer than normal cells (a sign the bacteria may have weak cell walls). Further experiments revealed that the cell walls of these bacteria were weaker than normal cells due to a shortage in a cell wall building material known as undecaprenyl phosphate.

Dewachter et al. then demonstrated that combining an existing drug known as clomiphene – which is known to inhibit undecaprenyl phosphate production and is currently used to treat infertility in humans – together with amoxicillin is able to effectively kill *S. pneumoniae* that are resistant to amoxicillin alone. Clomiphene also boosted the activity of amoxicillin against *S. pneumoniae* that remain sensitive to the antibiotic.

Before this new drug combination may be used to help treat *S. pneumoniae* infections in human patients, further experiments will be needed to find out the optimum dose of clomiphene to use with amoxicillin. In the future, the new screening approach developed by Dewachter et al. may also prove useful to other researchers studying a wide range of biological questions.

## Introduction

*Streptococcus pneumoniae* (the pneumococcus) is an opportunistic human pathogen that is responsible for diseases such as pneumonia, middle ear infections, sepsis, and meningitis (*Weiser et al., 2018*; *O'Brien et al., 2009*; *Bogaert et al., 2004*). *S. pneumoniae* is listed by the WHO as a priority pathogen for the development of novel antibiotics since it is one of the leading causes of fatal bacterial infections worldwide (*Weiser et al., 2018*; *O'Brien et al., 2009*; *Jensen et al., 2015*), and antibiotic resistance is on the rise (*Càmara et al., 2018*; *Schrag et al., 2004*; *Hakenbeck et al., 2012*). The alarming increase in prevalence of penicillin non-susceptible *S. pneumoniae* was initially dealt with by the development and roll-out of the pneumococcal conjugate vaccines PCV7, PCV10, and PCV13 in 2001, 2009, and 2010, respectively (*Càmara et al., 2018*). These vaccines induce protection against infection caused by 7, 10, or 13 of the most prevalent serotypes of capsular polysaccharides that surround *S. pneumoniae* (*Bogaert et al., 2004*). Since many of the targeted serotypes are associated with penicillin non-susceptibility, the occurrence of infections with such resistant strains decreased at first (*Càmara et al., 2018*; *Kaur et al., 2016*; *Brueggemann et al., 2007*). However, due to the high genomic plasticity of *S. pneumoniae* (*Hanage et al., 2009*; *Domenech, 2020*), serotype switching created penicillin non-susceptible escape mutants that are not covered by the currently available vaccines and that are once again gaining in prevalence (*Càmara et al., 2018*; *Kaur et al., 2016*).

Penicillin non-susceptible *S. pneumoniae* strains carry a variety of mutations in their penicillin-binding proteins (PBPs) that decrease their affinity for penicillin and therefore increase resistance (*Jensen et al., 2015*; *Hakenbeck et al., 2012*; *Chi et al., 2007*). These mutations do not only lower the susceptibility to penicillin but also to other β-lactam antibiotics (*Prieto et al., 2002*). However, the efficacy of aminopenicillins such as amoxicillin is usually less affected by these mutations, meaning that amoxicillin remains a viable treatment option for many penicillin non-susceptible strains and therefore became one of the frontline antibiotics to treat pneumococcal infections (*Prieto et al., 2002*; *Baquero et al., 2002*; *Jacobs, 2004*; *Bradley et al., 2011*; *Thiem et al., 2011*). However, highly

penicillin-resistant strains have been found to also be amoxicillin non-susceptible (*Càmara et al., 2018*; *Schrag et al., 2004*), thereby now also threatening the clinical efficacy of this widely used antibiotic.

Because of the rise in antibiotic resistance, we urgently need new antibiotic targets and/or need to find ways to extend the clinical efficacy of our current antibiotic arsenal. To contribute toward this goal, we developed sCRilecs-seq (subsets of CRISPR interference libraries extracted by fluorescence activated cell sorting coupled to next generation sequencing), a high-throughput single-cell based screening approach that we exploited to find targets that could resensitize resistant *S. pneumoniae* strains toward amoxicillin. sCRilecs-seq is based upon genome-wide gene silencing using CRISPR interference (CRISPRi), which makes use of a catalytically dead form of the Cas9 enzyme, called dCas9 (*Qi et al., 2013*). dCas9 is guided toward its target site in the DNA by the provided sgRNA(single guide RNA) but is unable to introduce double-stranded breaks. Instead, the dCas9 enzyme acts as a roadblock for RNA polymerase, thereby halting transcription (*Qi et al., 2013*; *Peters et al., 2016*; *Bikard et al., 2013*). CRISPRi gene silencing not only affects the target gene but also influences the expression of an entire transcriptional unit (*Qi et al., 2013*; *Peters et al., 2016*; *Bikard et al., 2013*; *Liu, 2017*; *Cui, 2018*). This technology therefore works at the operon level and generates strong polar effects that need to be considered. Despite this limitation, CRISPRi has proven to be a very powerful genetic tool to perform genome-wide depletion screens in various bacteria (*Cui, 2018*; *Liu et al., 2021*; *de Wet et al., 2018*; *Lee et al., 2019*; *Rousset et al., 2018*; *Wang et al., 2018*). In the sCRi-lecs-seq screen used here, subpopulations that display a phenotype of interest are collected using fluorescence activated cell sorting (FACS), as has previously been done in eukaryotic cells (*Park et al., 2019*) and for bacterial transposon libraries (*Smith et al., 2021*). The abundance of sgRNAs in the sorted fractions is then compared to a defined control population to look for genetic factors involved in the phenotype of interest. This highly flexible screening approach allowed us to identify targets that can be exploited in our fight against bacterial infections.

Our sCRilecs-seq results highlight the importance of the mevalonate pathway for maintaining proper cell morphology in *S. pneumoniae*. The mevalonate pathway is the only pathway used by *S. pneumoniae* to produce isoprenoids, a highly diverse class of organic molecules that are essential to all life on earth and that function – among others – in processes such as cell wall synthesis, electron transport, membrane stability, and protein modification (*Boronat and Rodríguez-Concepción, 2015*; *Heuston et al., 2012*). Our screen revealed that inhibition of the mevalonate pathway in *S. pneumoniae* leads to extensive cell elongation. The data suggest that this elongation is caused by a deficiency in peptidoglycan precursors at the site of cell wall synthesis, which is in turn due to a limitation in the production of undecaprenyl phosphate (Und-P), the lipid carrier that is responsible for transporting these precursors across the cell membrane (*Bouhss et al., 2008*). This shortage of peptidoglycan precursors allows cell elongation to continue but causes a block in cell division.

Additionally, based on the mevalonate depletion phenotype characterized here, we successfully designed a combination treatment strategy targeting *S. pneumoniae*. In this combination treatment strategy, amoxicillin is potentiated by the simultaneous administration of clomiphene, a widely used FDA-approved fertility drug that was shown to block Und-P production in *Staphylococcus aureus* and to synergize with β-lactam antibiotics (*Farha et al., 2015*). This combination of compounds is particularly powerful against amoxicillin-resistant *S. pneumoniae* strains, as it is capable of resensitizing these strains to clinically relevant concentrations of amoxicillin in vitro. Our findings could therefore provide a useful starting point for the development of treatment strategies that are effective against the rising amount of amoxicillin-resistant *S. pneumoniae* infections and could extend the clinical efficacy of this important antibiotic.

## Results

### Amoxicillin treated *S. pneumoniae* is elongated

To identify new druggable pathways that could potentiate the antimicrobial activity of amoxicillin, we first established the impact of amoxicillin on pneumococcal morphology and growth. We therefore performed time-lapse microscopy of *S. pneumoniae* in the presence or absence of amoxicillin. As shown in *Figure 1A–B* and *Videos 1–2*, amoxicillin-treated cells initially elongate before they lyse. Indeed, quantitative image analysis confirmed that amoxicillin, like some other β-lactam antibiotics

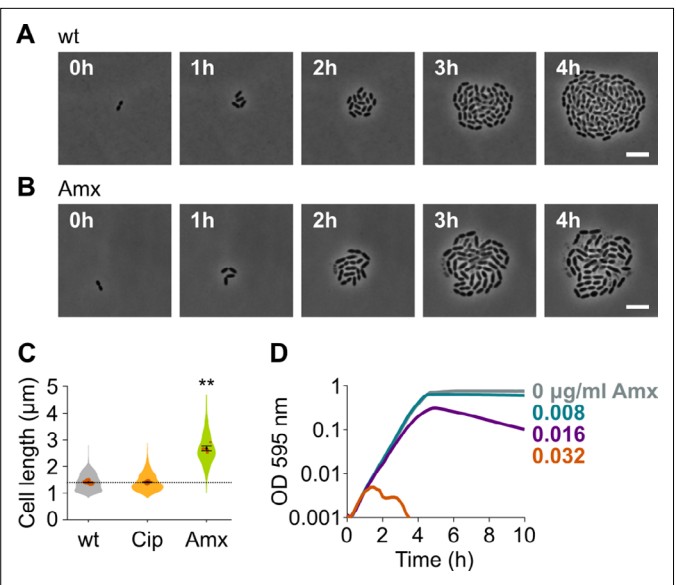

**Figure 1.** Amoxicillin causes cell elongation before triggering cell lysis. (**A**) Snapshots of a time lapse phase contrast microscopy experiment of *S. pneumoniae* D39V growing in the absence of antibiotics (*Video 1*). (**B**) Snapshots of a time-lapse analysis of *S. pneumoniae* D39V growing in the presence of a sub-MIC concentration of amoxicillin (0.016 µg/ml) (*Video 2*). (**C**) The effect of sub-MIC concentrations of ciprofloxacin (0.5 µg/ml) and amoxicillin (0.016 µg/ml) on the cell length of *S. pneumoniae* D39V was tested by phase contrast microscopy after 2 hr of treatment. Quantitative analysis of micrographs shows that cell length increases upon treatment with amoxicillin. Data are represented as violin plots with the mean cell length of every biological repeat indicated with orange dots. The size of these dots indicates the number of cells recorded in each repeat, ranging from 112 to 1498 cells. Black dots represent the mean ± SEM of all recorded means, n≥3. Two-sided Wilcoxon signed rank tests were performed against wt without antibiotic as control group (dotted line), and p values were adjusted with a false discovery rate (FDR) correction; ** p<0.01. (**D**) *S. pneumoniae* D39V was grown in the presence of different concentrations of amoxicillin, and growth was followed by monitoring OD 595 nm. wt: wildtype; Cip: ciprofloxacin; Amx: amoxicillin.

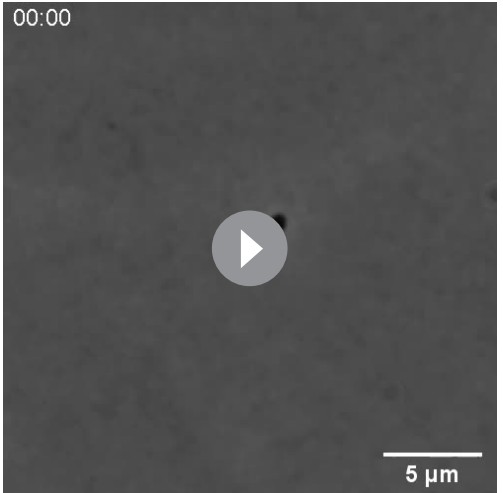

**Video 1.** *Streptococcus pneumoniae* D39V (VL333) growing on agarose pads of C+Y medium without added compounds.

https://elifesciences.org/articles/75607/figures#video1

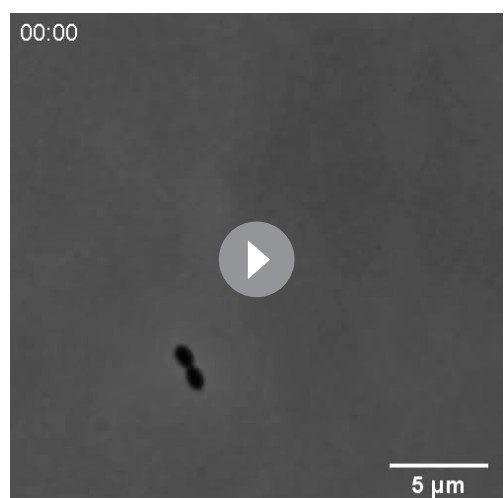

**Video 2.** *Streptococcus pneumoniae* D39V (VL333) growing on agarose pads of C+Y medium supplemented with 0.016 µg/ml amoxicillin.

https://elifesciences.org/articles/75607/figures#video2

(*Tsui et al., 2014*), causes cell elongation (*Figure 1C*), while growth curves in the presence of this antibiotic demonstrate that lysis occurs since optical density (OD) decreases at high amoxicillin concentrations (*Figure 1D*). To demonstrate that cell elongation is specifically caused by amoxicillin and does not generally occur upon antibiotic treatment, we also assessed cell lengths in the presence of the fluoroquinolone ciprofloxacin. As expected, ciprofloxacin did not induce cell elongation (*Sorg and Veening, 2015*; *Figure 1C*). Because it was shown that synergy between antimicrobial compounds is most often detected when they target the same process (*Brochado et al., 2018*), we hypothesized that targets whose inhibition would cause cell elongation, like amoxicillin, could potentially reinforce the antibacterial effect of this antibiotic.

## Development of sCRilecs-seq

To identify pathways that lead to cell elongation upon inhibition, we developed a single-cell based, high-throughput screening strategy where we combined CRISPRi gene silencing with high-throughput selection of a phenotype of interest by FACS. Using this approach, which we call sCRilecs-seq, we screened for mutants that display an increase in forward scatter (FSC), which is a rough indicator of cell size (*Shi et al., 2017*). To facilitate downstream analyses, screening was performed in an *S. pneumoniae* strain in which the DNA and the cell division protein FtsZ are marked. DNA was visualized using the DNA-binding protein HlpA fused to GFP. HlpA, also known as HU, is a histone-like protein that binds DNA aspecifically and is highly expressed in *S. pneumoniae* (*Kjos et al., 2015*). It is therefore often used as a nucleoid marker (*Kjos et al., 2015*; *Beilharz et al., 2015*). FtsZ was marked by fusing it to mCherry. To this end, the native *hlpA* and *ftsZ* genes of *S. pneumoniae* D39V were replaced by *hlpA-gfp* and *ftsZ-mCherry*, respectively. Both fusions were previously shown to support viability (*Kjos et al., 2015*; *Beilharz et al., 2015*). Additionally, the gene encoding dCas9 was inserted into the pneumococcal chromosome and was placed under tight control of an IPTG-inducible $P_{lac}$ promoter (*Liu, 2017*; *Liu et al., 2021*). The resulting strain (VL3117: D39V *lacI* $P_{lac}$-*dcas9 hlpA-gfp ftsZ-mCherry*) was transformed with a pool of 1499 different integrative plasmids carrying constitutively expressed sgRNAs (under control of the P3 promoter) that together target the entire coding genome (*Liu et al., 2021*). This way, we created a pooled CRISPRi library where each cell expresses a certain sgRNA resulting in transcriptional downregulation of a specific gene or operon upon induction of dCas9 (*Figure 2A*). Note that sgRNAs were designed in such a way that the chance for off targeting effects is minimal (*Liu et al., 2021*).

We next grew the library in C+Y medium at 37°C for 3.5 hr in the presence of 1 mM IPTG to induce dCas9. The pooled and induced library was subjected to FACS and different fractions of the population were sorted (see Materials and methods). Of note, *S. pneumoniae* grows as short cell chains (*Massidda et al., 2013*) and the number of cells present in a chain could influence the FSC read-out. To ensure that measurements were taken at the single-cell level, chains were mechanically disrupted by vigorous vortexing before sorting took place. We confirmed that this approach is successful at breaking up cell chains ensuring that morphology was evaluated for single cells rather than entire chains (*Figure 2—figure supplement 1A-C*). 10% of the population with the highest FSC values was collected. As a control, the centermost 70% of the population was sorted as well. For both collected fractions, sgRNAs were amplified by PCR using primers that contain Illumina adapters (see Materials and methods). After sequencing, sgRNA read counts were compared between the sorted fractions. This approach (*Figure 2A*) enabled us to identify sgRNAs that are significantly enriched in the fraction of the population with highest FSC values. These sgRNAs point to genes or operons necessary to maintain normal cell size.

## sCRilecs-seq identifies several targets involved in maintaining proper cell size

Following this strategy, we were able to identify 70 sgRNAs that were significantly enriched in the fraction of the population with the highest FSC values (*Figure 2B*, *Figure 2—figure supplement 1D* and *Supplementary file 1*). To validate whether the identified sgRNAs represent true hits, we selected eight sgRNAs that were found to be among the most highly enriched (highest log2 fold change [log2FC]). These sgRNAs were individually cloned and transformed to strain D39V $P_{lac}$-*dcas9 lacI hlpA-gfp ftsZ-mCherry* (VL3117), and their effect was evaluated by flow cytometry (*Figure 2C*) and microscopy (*Figure 2D* and *Figure 2—figure supplement 1E*). These results demonstrate that

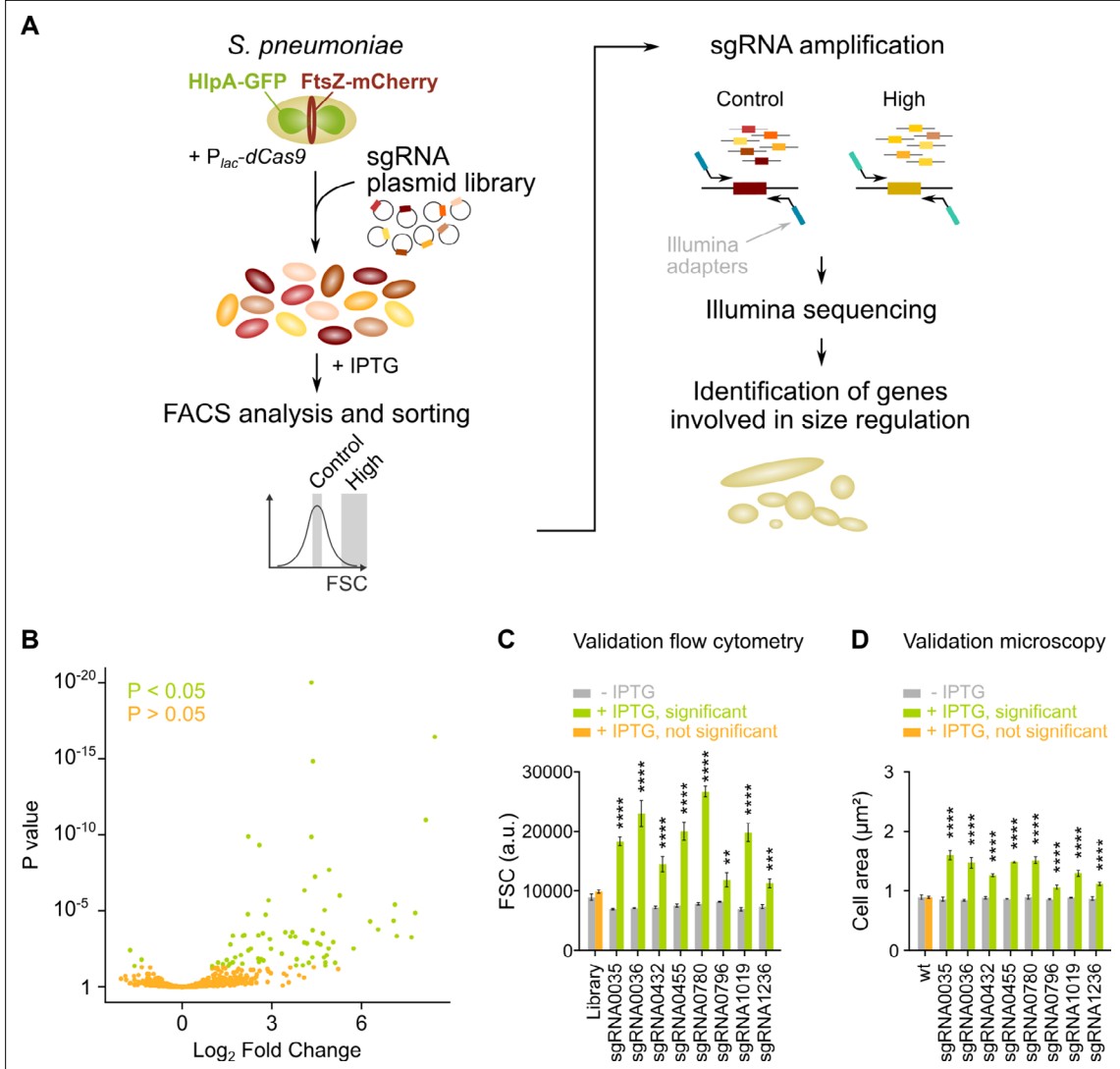

**Figure 2.** sCRilecs-seq (subsets of CRISPR interference libraries extracted by fluorescence activated cell sorting coupled to next generation sequencing) identifies operons involved in cell size regulation. (**A**) A pooled CRISPR interference (CRISPRi) library was constructed in *S. pneumoniae* D39V *P_lac-dcas9 lacI hlpA-gfp ftsZ-mCherry* (VL3117) by transformation of a plasmid library encoding 1499 constitutively expressed sgRNAs that together target the entire genome. This CRISPRi library was grown in the presence of IPTG for 3.5 hr to induce expression of dCas9, and cultures were sorted based on forward scatter (FSC) as a proxy for cell size. 10% of the population with the highest FSC values was sorted, as well as the centermost 70% of the population which served as a control. sgRNAs from sorted fractions were amplified by PCR using primers that contain Illumina adapters. Amplified sgRNAs were sequenced and mapped to the sgRNA library. sgRNA read counts were compared between the different sorted fractions to identify gene depletions that lead to increases in cell size. (**B**) A volcano plot shows the statistical significance and enrichment for every sgRNA in the fraction of the population with high FSC values compared to the control with normal FSC values. (**C–D**) Some of the most strongly enriched significant sgRNA hits were validated by studying individual mutants. (**C**) Flow cytometry measurements of mutants grown with and without IPTG were performed. The median FSC value for each depletion was recorded and compared to the median value of the same strain without induction of dCas9. Note that the entire CRISPRi library was also included in this experiment ('Library') as a control. Data are represented as mean ± SEM, n=3. ** p<0.01, *** p<0.001, **** p<0.0001, and two-sided t-tests with Holm-Sidak correction for multiple comparisons. (**D**) Quantitative analysis of microscopy images of CRISPRi depletion mutants with and without IPTG was performed. The mean of the mean cell area for each depletion was measured and compared to the same strain without induction of dCas9. Data are represented as mean of mean ± SEM, n=3 where each repeat consists of at least 50 cells. **** p<0.0001 and two-sided Wilcoxon signed rank tests with false discovery rate (FDR) correction for multiple comparisons.

The online version of this article includes the following figure supplement(s) for figure 2:

**Figure supplement 1.** The sCRilecs-seq (subsets of CRISPR interference libraries extracted by fluorescence activated cell sorting coupled to next generation sequencing) screen started from mechanically separated single cells and displayed a high amount of variation.

all selected hits from the sCRilecs-seq screen led to increased FSC values and larger cell sizes upon depletion, meaning that all sgRNAs selected for validation represent true positive hits. We therefore conclude that our screen is capable of reliably detecting genes or operons that upon repression lead to increased cell size.

The reliability of our screening approach is also reflected by the fact that many of the significant hits are known to be involved in processes that influence cell size and/or morphology (*Supplementary file 1*). Such known regulators that are picked up by our screen include the late cell division proteins DivIB, DivIC, FtsL, and FtsW (*Massidda et al., 2013*) and the cell cycle regulator GpsB (*Rued et al., 2017*; *Land et al., 2013*). Also, when performing a gene ontology (GO) enrichment analysis (*Ashburner et al., 2000*; *Mi et al., 2019*; *Consortium, 2019*) on these hits, we detect categories that are expected to play an important role in determining cell size (*Supplementary file 2*). Biological processes found to be enriched include 'regulation of cell shape' (fold change 8.43, p value 1.03E-05), 'cell wall organization' (fold change 6.71, p value 7.24E-04), 'cell division' (fold change 6.63, p value 1.24E-04), and 'peptidoglycan biosynthetic process' (fold change 6.27, p value 2.26E-03). Besides these processes, it is also notable that many hits target the production of teichoic acids. These glyco-polymers are an important component of the *S. pneumoniae* cell envelope and are indeed necessary to maintain proper cell morphology (*Liu, 2017*; *Vollmer et al., 2019*; *Brown et al., 2013*). Our list of hits also contains several components of the Sec translocase system, thereby confirming the previously published link between this translocase and *S. pneumoniae* cell division (*Tsui et al., 2011*). Additional hits include the operon encoding the VicRK two-component system that is thought to control cell size by regulating the expression of the murine hydrolase PcsB that is also detected as a significant hit in our screen and depletion of which is known to lead to uncontrolled peptidoglycan synthesis and large cells (*Ng et al., 2004*).

Surprisingly, among hits associated with increased cell size, we also find some genes that are expected to lead to a decrease in cell length upon depletion. These genes include *mreD* (*Land and Winkler, 2011*), *divIVA* (*Fleurie et al., 2014*), *pbp2b* (*Tsui et al., 2014*; *Berg et al., 2013*), and *rodA* (*Massidda et al., 2013*). To investigate why these gene depletions expected to be associated with decreases in cell length are detected among cells with the highest FSC values, we have constructed individual CRISPRi depletion strains of these genes and have checked their effect on cell size using microscopy. Results indeed confirm that both cell width and cell length are increased for CRISPRi depletion of operons containing *mreD*, *pbp2b,* and *rodA* (*Figure 2—figure supplement 1F-H*). We think these unexpected results can either be explained by operon effects or are the consequence of severe cell swelling preceding lysis.

Additionally, depletion of expression of several genes encoding hypothetical proteins (i.e. SPV_0010, SPV_0131, SPV_0418, operon SPV_1594/SPV_1595, and SPV_1931) led to enlarged cells. Our results clearly point toward a role for these genes in cell size regulation which could potentially occur through an effect on cell division or the cell envelope. Future research should be able to build upon these first clues and further unravel the cellular role of these unknown proteins.

Finally, whereas the GO enrichment analysis rendered a lot of expected and anticipated results, the most highly upregulated category was 'isoprenoid biosynthetic process' (fold change 13.43, p value 6.45E-03). We therefore decided to focus on isoprenoid biosynthesis and how disturbances in this process lead to changes in *S. pneumoniae* cell size that might be exploited as a novel antimicrobial strategy.

## Depletion of mevalonate pathway components leads to cell elongation

*S. pneumoniae* synthesizes isoprenoids through the mevalonate pathway (*Boronat and Rodríguez-Concepción, 2015*; *Wilding et al., 2000*). This pathway, with mevalonic acid (mevalonate) as intermediate, produces the C5 precursors, isopentenyl diphosphate, and dimethylallyl diphosphate (*Figure 3A*), which can be condensed into large and diverse isoprenoids (*Boronat and Rodríguez-Concepción, 2015*). In *S. pneumoniae*, all proteins involved in the mevalonate pathway are encoded by two different operons (*Wilding et al., 2000*; *Slager et al., 2018*), both of which were identified as hits in our sCRilecs-seq screen. In a first step, we made non-CRISPRi-based deletion/complementation strains for both operons. In an *S. pneumoniae* D39V strain that encodes the *hlpA-gfp* and *ftsZ-mCherry* fusion proteins as well as LacI (VL3404), complementation constructs for the mevalonate genes under control of the IPTG-inducible P$_{lac}$ promoter were inserted ectopically into the genome,

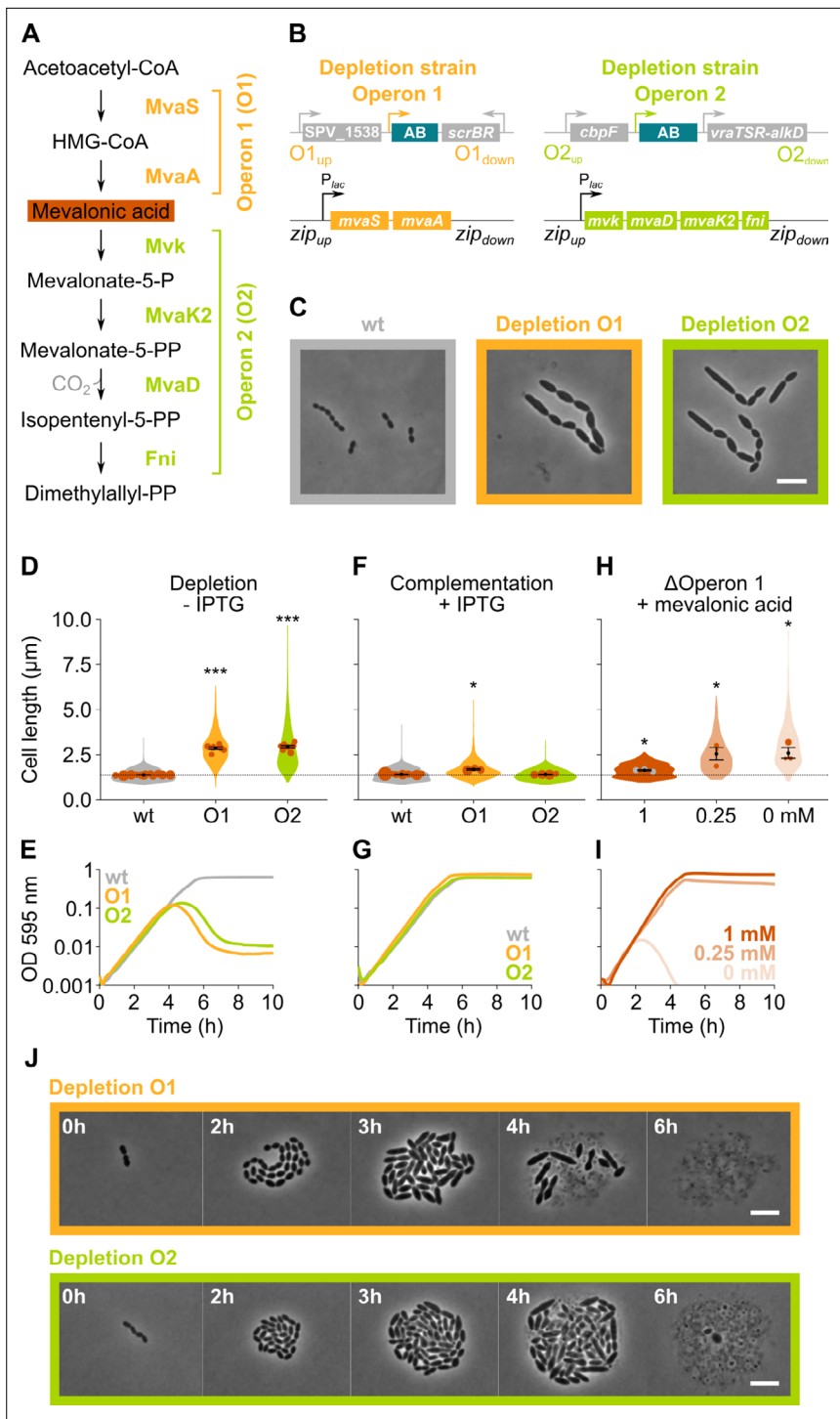

**Figure 3.** The mevalonate pathway is essential for *S. pneumoniae* and leads to cell elongation upon depletion. (**A**) The mevalonate pathway and its genetic organization in *S. pneumoniae* is depicted. (**B**) A genetic representation of the mevalonate depletion strains are shown. The native mevalonate operons were deleted and replaced by an antibiotic marker (AB), and a complementation construct under control of the P$_{lac}$ promoter was inserted at the *zip* locus (*Keller et al., 2019*) in the *S. pneumoniae* genome. (**C**) Phase contrast microscopy images of liquid cultures of *S. pneumoniae* wildtype (wt) or upon depletion of one of the mevalonate operons for 4 hr in VL3565 and VL3567 are shown. Scale bar, 5 μm. (**D**) Quantitative analysis of phase contrast micrographs shows that cell length increased when mevalonate operons were depleted. Data are represented as violin plots with the mean cell length of every biological repeat indicated with orange dots. The size of these dots indicates the

*Figure 3 continued on next page*

*Figure 3 continued*

number of cells recorded in each repeat, ranging from 100 to 2626 cells. Black dots represent the mean ± SEM of all recorded means, n≥3. (**E**) Depletion of mevalonate operons led to a severe growth defect. Data are represented as the mean, n≥3. (**F**) The elongated phenotype upon depletion of mevalonate operons could be complemented by inducing their expression with IPTG. Data are represented as violin plots with the mean cell length of every biological repeat indicated with orange dots. The size of these dots indicates the number of cells recorded in each repeat, ranging from 100 to 2626 cells. Black dots represent the mean ± SEM of all recorded means, n≥3. (**G**) The growth defect associated with depletion of mevalonate operons could be fully complemented by inducing their expression with IPTG. Data are represented as the mean, n≥3. (**H**) A mutant in which the first mevalonate operon was deleted (VL3702, no complementation construct) displayed increased cell length when the concentration of mevalonic acid added to the growth medium was decreased. Data are represented as violin plots with the mean cell length of every biological repeat indicated with orange (or gray) dots. The size of these dots indicates the number of cells recorded in each repeat, ranging from 100 to 2626 cells. Black dots represent the mean ± SEM of all recorded means, n≥3. (**I**) Growing the mutant in which the first mevalonate operon is deleted (no complementation construct) with decreasing concentrations of external mevalonic acid led to an increasing growth defect, resulting in full extinction of the culture when no mevalonic acid was provided. Data are represented as the mean, n≥3. (**J**) Snapshot images of phase contrast time lapse experiments with mutants in which the first or second mevalonate operon was depleted (VL3565 and VL3567) are shown. Strains were grown on agarose pads of C+Y medium without the inducer IPTG. Scale bar, 5 µm. Two-sided Wilcoxon signed rank tests were performed against wt – IPTG as control group (dotted line) and p values were adjusted with a false discovery rate (FDR) correction; * p<0.05 and *** p<0.001.

The online version of this article includes the following figure supplement(s) for figure 3:

**Figure supplement 1.** The mevalonate pathway is essential for *S. pneumoniae* and leads to cell elongation upon depletion.

**Figure supplement 2.** Suppressor mutants arise due to mutations in *lacI* or P*lac*.

and the native mevalonate genes were deleted (***Figure 3B***). The resulting two deletion/complementation strains (VL3565 and VL3567) were then used to assess the effect of mevalonate synthesis on cell size and growth.

Phase contrast microscopy of cells depleted for the mevalonate pathway operons demonstrated large increases in cell length (***Figure 3C–D***), confirming the sCRilecs-seq screen. Even though cells also became wider upon mevalonate depletion (***Figure 3C***), cell length was disproportionally affected. Cells on average attained more than twice their normal length when the mevalonate operons were depleted (***Figure 3D***), whereas the mean cell width increased 1.35× for depletion of operon 1 and 1.25× for operon 2. The disproportionately large effect on cell elongation is also demonstrated by significant increases in the length/width ratio (***Figure 3—figure supplement 1A***). In addition to cell elongation, depletion of the mevalonate operons led to severe growth defects (***Figure 3E***). Importantly, the elongated phenotype and observed growth defects could be complemented by ectopic induction of the mevalonate operons with an optimized concentration of IPTG (***Figure 3F–G***), demonstrating that these phenotypes are not due to polar effects of the mutations. Time-lapse microscopy upon depletion of the mevalonate operons confirmed these findings, showing that cells initially continue to grow and elongate before lysis takes place (***Figure 3J*** and ***Videos 2–3***). Of note, lysis was found to be partially dependent on LytA, and phenotypes were reproducible in deletion/complementation strains that do not carry the *hlpA-gfp* and *ftsZ-mCherry* fusions (VL3708 and VL3709) (***Figure 3—figure supplement 1B-G***).

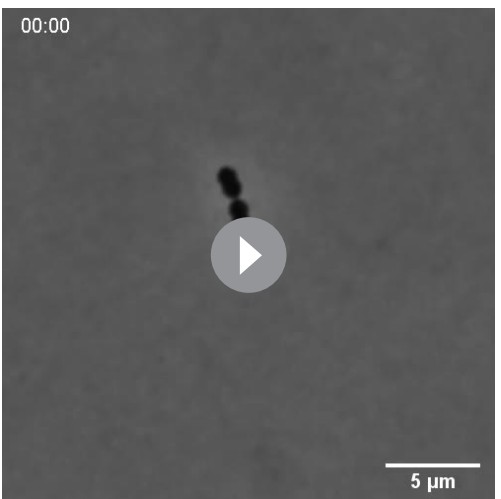

**Video 3.** An *S. pneumoniae* mutant strain where the first mevalonate operon was depleted (VL3709) growing on agarose pads of C+Y medium without the inducer IPTG.

https://elifesciences.org/articles/75607/figures#video3

Although the severe growth defects associated with depletion of the genes involved in mevalonate biosynthesis point to the importance of this pathway for *S. pneumoniae* physiology, cultures were not fully sterilized at late time points and many survivors remained (*Figure 3E* and *Figure 3—figure supplement 2A*). To test whether the mevalonate pathway is essential for *S. pneumoniae* viability and that the survival of the depletion strains at late time points is due to leaky expression and/or suppressor mutations, we attempted to delete the mevalonate operons without providing a complementation construct. As expected for essential genes, we were unable to delete these operons. However, when mevalonic acid – an intermediary in the mevalonate pathway (*Figure 3A*) – was added to the growth medium, we were able to delete operon 1, as was also shown previously in *S. pneumoniae* and *S. aureus* (*Wilding et al., 2000*; *Balibar et al., 2009*; *Yu et al., 2013*; *Reichert et al., 2018*). As expected, based on pathway architecture, deleting operon 2 – which encodes proteins that work downstream of mevalonic acid – remained impossible. We next grew *S. pneumoniae Δoperon 1* (VL3702) with different concentrations of mevalonic acid. Cells grew normally and attained normal cell lengths when a high concentration of mevalonic acid was added to the growth medium (*Figure 3H–I*). However, decreasing the concentration of mevalonic acid led to cell elongation and growth defects that became more pronounced at lower concentrations, thereby phenocopying genetic depletion of operon 1 and 2. In fact, when no mevalonic acid is provided at all, cultures are driven to full extinction (*Figure 3I* and *Figure 3—figure supplement 2B*). These results confirm that the mevalonate pathway is essential for *S. pneumoniae* viability and that it is the only pathway through which this bacterium can synthesize isoprenoids (*Wilding et al., 2000*). The differences in growth and lysis observed between the deletion and depletion strains can be attributed to the presence of suppressor mutations in the latter that allow for IPTG-independent expression of the mevalonate operons (*Figure 3—figure supplement 2C-G*).

## Depletion of mevalonate operons primarily interferes with cell division

We next asked how depletion of mevalonate pathway components results in an elongated phenotype. Microscopic investigation of mevalonate depleted cells showed that these bacteria contain many unconstricted FtsZ rings (*Figure 4A–B*). High-resolution investigation of this phenotype by transmission electron microscopy (TEM) showed that depletion of the mevalonate operons resulted in an unusually high occurrence of initiated septa that appear to be blocked for further constriction, whereas division septa at all stages of constriction could be found in a wild-type (wt) strain (*Figure 4C*). The inability to constrict and divide could explain the observed filamentation if cell elongation still occurs.

To investigate if indeed peptidoglycan synthesis for cell elongation remains active upon depletion of the mevalonate operons, we labeled sites of active peptidoglycan synthesis with fluorescent D-amino acids (FDAAs) (*Kuru et al., 2015*). In a first pulse, cells were labeled with the green FDAA, sBADA, for 15 min, and a second 15 min pulse consisted of the red FDAA, RADA (see Materials and methods). Fluorescence microscopy showed that peptidoglycan synthesis leads to both cell elongation and division in a wt strain (*Figure 4D*). Likewise, also the mevalonate depletion strains showed sites of active peptidoglycan synthesis, indicating that not all cell wall production is halted. However, in this case, almost no constricted FDAA-labeled sites were found (*Figure 4D*). Indeed, when defining septal peptidoglycan synthesis as FDAA bands that are less than 70% of the maximal cell width (i.e. sites of constriction, a criterion also used previously *Wheeler et al., 2011*), quantitative analysis of hundreds of peptidoglycan synthesis sites demonstrated that the amount of septal synthesis is drastically reduced when transcription of the mevalonate operons is repressed (*Figure 4E*). Additionally, even though peptidoglycan synthesis continues upon mevalonate depletion, this process is slowed down since FDAA intensity is decreased under these conditions, indicating that less peptidoglycan is being produced (*Figure 4D*). Quantification of the intensity of the sBADA signal indeed confirmed that the incorporation of the label is strongly decreased upon mevalonate depletion (*Figure 4F*). The same quantification was not done for the RADA label since the relatively high aspecific background fluorescence interfered strongly with this analysis.

Taken together, these results strongly indicate that mevalonate depletion leads to filamentation because constriction and cell division are blocked while peptidoglycan synthesis that contributes to cell elongation can still occur, albeit at a lower rate. However, elongation does not continue indefinitely and cells eventually lyse (*Figure 3J* and *Videos 3–4*). Since lysis usually occurs through weakening of the peptidoglycan cell wall (*Yao et al., 2012*), we hypothesize that extended depletion of mevalonate

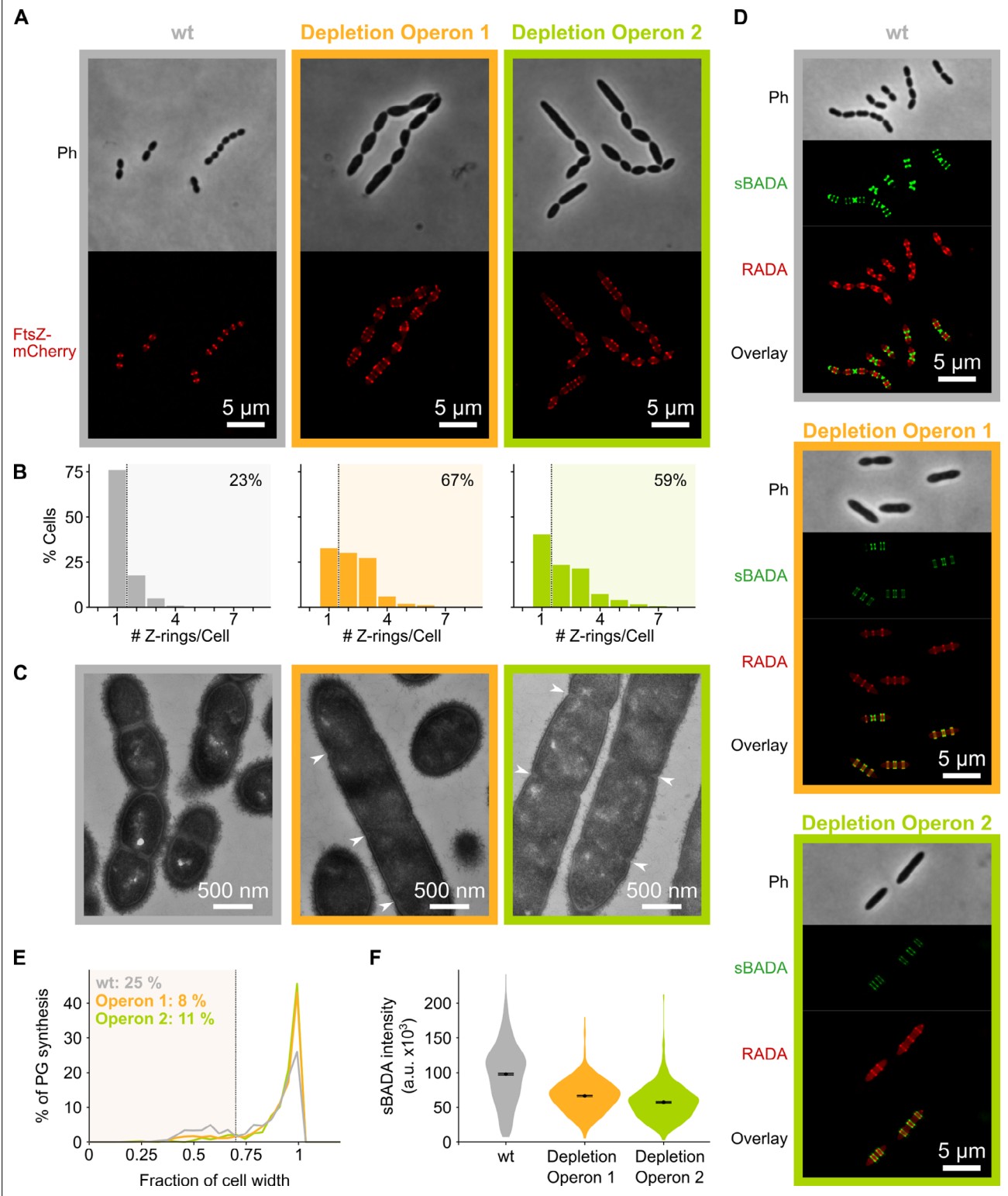

**Figure 4.** Depletion of mevalonate operons prevents cell division. (**A**) While *S. pneumoniae* wild-type (wt) cells typically contain one FtsZ ring at the cell center, depletion of either one of the mevalonate operons led to strongly elongated cells with multiple unconstricted FtsZ rings. Images were obtained using strains VL3404, VL3565, and VL3567 that encode the *ftsZ-mCherry* fusion. (**B**) Quantitative analysis of microscopy pictures was used to determine the number of Z-rings per cell. Pictures from at least six independent repeats were analyzed, each including over 100 cells. (**C**) Transmission electron microscopy (TEM) images show that elongated cells contained many initiated septa that appear to be blocked in further progression of constriction (white arrow heads). Images were obtained using strains VL333, VL3708, and VL3709 that do not encode fluorescent fusion proteins. (**D**) Pulse labeling *S.*

*Figure 4 continued on next page*

*Figure 4 continued*

*pneumoniae* with the green fluorescent D-amino acid (FDAA), sBADA, and 15 min later with the red RADA shows sites of active peptidoglycan synthesis involved in either elongation or constriction. Images were obtained using strains VL333, VL3708, and VL3709 that do not encode fluorescent fusion proteins. In overlay images, sBADA and RADA intensities were freely adjusted to produce the clearest images. Intensities in individual channels were not manipulated. (**E–F**) Quantitative image analysis of sites of active peptidoglycan synthesis labeled with FDAAs shows that depletion of mevalonate operons eliminated septal peptidoglycan synthesis since virtually no narrow sBADA or RADA bands can be found (**E**) and that the intensity of FDAA labeling decreased upon mevalonate depletion, indicating that peripheral peptidoglycan synthesis was slowed down (**F**). sBADA intensity is represented as the total intensity per cell. Number of sBADA bands analyzed for each condition >400.

operons eventually affects the integrity and/or rigidity of the peptidoglycan mesh, leading to cell lysis. However, considering the initial phenotypes found, we conclude that the primary effect of mevalonate depletion is a block in constriction necessary for cell division.

## Depletion of mevalonate operons prevents cell division by decreasing the amount of peptidoglycan precursors available for cell wall synthesis

The mevalonate pathway provides the precursor for the production of all isoprenoids in *S. pneumoniae* (*Boronat and Rodríguez-Concepción, 2015*; *Wilding et al., 2000*). Thus, a shortage in the production of any of these isoprenoids could underlie the phenotypic effects of perturbed mevalonate production. However, given the nature of the observed phenotype, we thought it most likely that a shortage in the production of Und-P would be responsible for the observed effects. This hypothesis is also supported by the fact that UppP, the gene product that dephosphorylates undecaprenyl pyrophosphate (Und-PP) to create Und-P, is found as a hit that increases cell size in our sCRilecs-seq screen (*Supplementary file 1*). Und-P is a C55 isoprenoid that acts as the lipid carrier for transport of cell envelope precursors from the cytoplasm to the extracellular environment (*Bouhss et al., 2008*). Und-P is produced by the dephosphorylation of Und-PP, which is in turn synthesized by UppS (undecaprenyl pyrophosphate synthase) by the addition of eight isopentenyl units to the C15 isoprenoid farnesyl-PP (*Boronat and Rodríguez-Concepción, 2015*; *Bouhss et al., 2008*; *Figure 5A*). To test whether the observed effects of mevalonate depletion are indeed caused by a deficiency of Und-P, we constructed a *uppS* deletion/complementation strain (VL3584, *Figure 5B*). When this strain was grown in absence of *uppS* expression, cells were elongated and growth was reduced (*Figure 5C–D*), thereby phenocopying mevalonate mutants. In addition, UppS depleted cells contained multiple unconstricted Z-rings and initiated septa that failed to constrict further, even though peptidoglycan synthesis still occurred as demonstrated by TEM and fluorescence microscopy (*Figure 5D–F*). Note that *uppS* was not identified by our sCRilecs-seq screen, likely because it is in an operon with several other essential genes not directly involved in cell division, such as *cdsA* and *proS* (*Slager et al., 2018*). We thus conclude that the negative effects caused by depletion of the mevalonate operons can be explained by insufficient production of Und-PP and subsequently Und-P.

The isoprenoid Und-P carries precursors of the capsule, teichoic acids, and peptidoglycan across the membrane (*Bouhss et al., 2008*). The observed phenotype of mevalonate and *uppS* mutants could therefore be caused by a shortage in either one or a combination of these components. To test which of these pathways contribute to the observed phenotypes upon mevalonate depletion, we tested each of them individually. First, we deleted the *cps* operon responsible for capsule synthesis in our D39V strain (*Yother, 2011*). As shown in *Figure 5—figure supplement 1A-B*, *cps* mutant cells were not elongated nor was growth rate affected, as also shown before

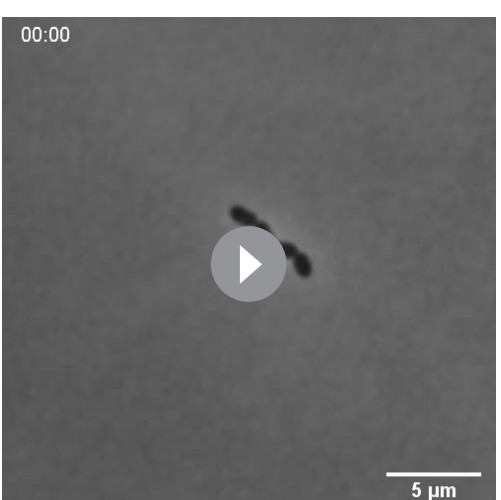

**Video 4.** An *S. pneumoniae* mutant strain where the second mevalonate operon was depleted (VL3708) growing on agarose pads of C+Y medium without the inducer IPTG.

https://elifesciences.org/articles/75607/figures#video4

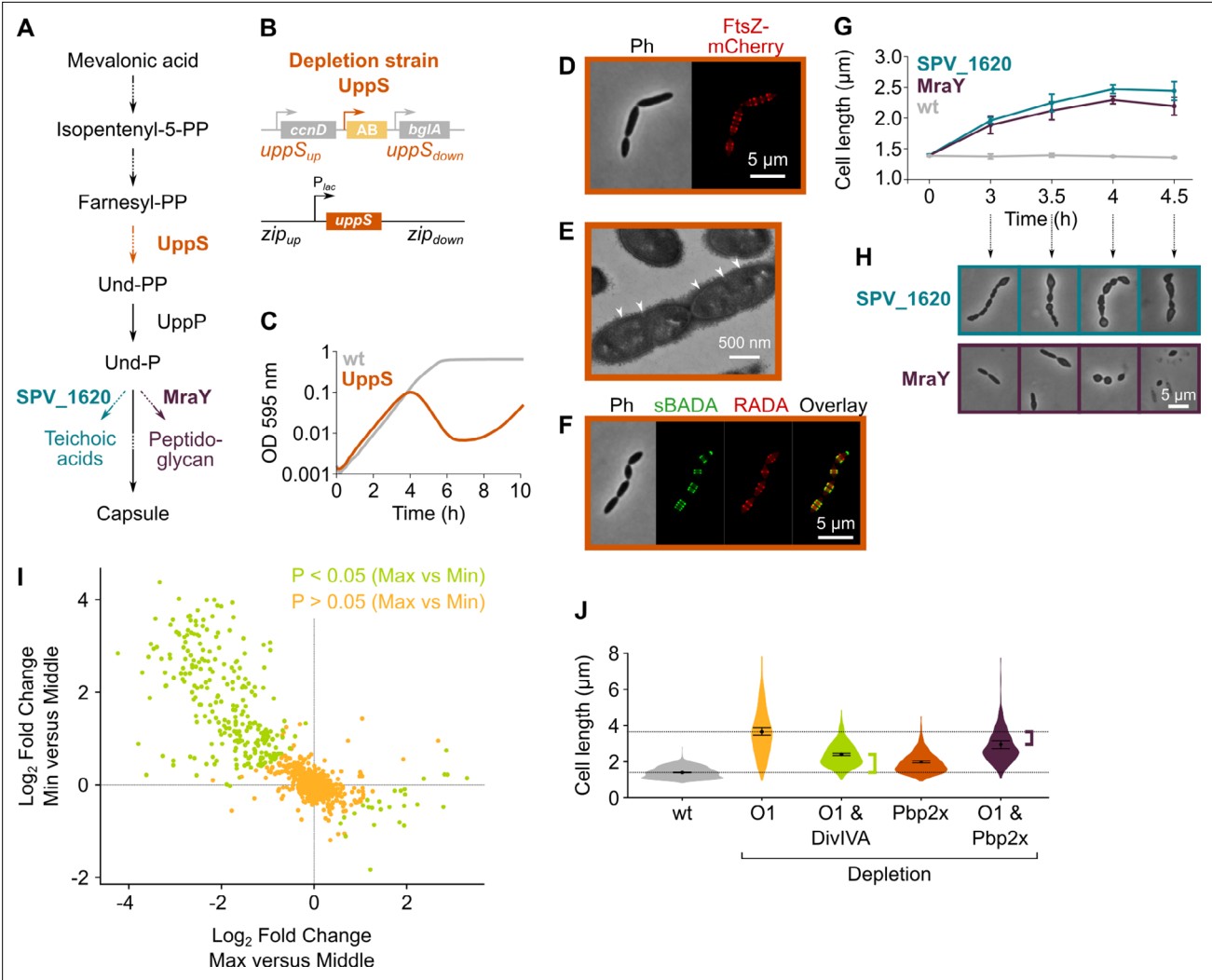

**Figure 5.** Depletion of the mevalonate pathway likely prevents cell division by decreasing the amount of peptidoglycan precursors available for cell wall synthesis. (**A**) After conversion of mevalonic acid into the basic isoprenoid building block isopentenyl-5-PP in the mevalonate pathway, this building block is condensed into the C15 molecule farnesyl-PP. Farnesyl-PP can be used by undecaprenyl pyrophosphate synthase (UppS) for the production of undecaprenyl pyrophosphate (Und-PP), which after dephosphorylation to undecaprenyl phosphate (Und-P) by UppP acts as the lipid carrier for the transport of precursors of peptidoglycan, the capsule, and teichoic acids across the cell membrane. (**B**) A genetic overview of the UppS depletion strains is shown. The native *uppS* gene was replaced by an antibiotic marker (AB), and a complementation construct under control of the $P_{lac}$ promoter was inserted at the *zip* locus in the *S. pneumoniae* genome. (**C**) Depletion of UppS in VL3584 caused a growth defect similar to depletion of the mevalonate operons. We confirmed that the growth observed after an initial phase of lysis was due to suppressor mutants that are no longer sensitive to UppS depletion (*Figure 5—figure supplement 1C*). Data are represented as averages, n≥3. (**D**) Like depletion of the mevalonate operons, depletion of UppS caused an elongated phenotype where cells contained multiple unconstricted FtsZ rings. Images were obtained using strain VL3585 which encodes *ftsZ-mCherry*. (**E**) Transmission electron microscopy (TEM) images show that cells elongated due to UppS depletion contained many initiated septa that appeared to be blocked in further progression of constriction (white arrow heads). Images were obtained using strain VL3710 that does not encode fluorescent fusion proteins. (**F**) Pulse labeling *S. pneumoniae* depleted for UppS with the green FDAA, sBADA, and subsequently with the red RADA dye showed sites of active peptidoglycan synthesis, which are in this case all directed at cell elongation. Images were obtained using strain VL3710 that does not encode fluorescent fusion proteins. In the overlay, sBADA and RADA intensities were freely adjusted to produce the clearest image. Intensities in individual channels were not manipulated. (**G**) The effect of the depletion of MraY and SPV_1620 on cell length was followed through time using quantitative image analysis. For each biological repeat (n≥3), more than 50 cells were used to calculate the average cell length. Data are represented as the mean ± SEM of these averages. (**H**) Representative morphologies of VL3585 and VL3586 corresponding to the analysis from panel G are shown. (**I**) A pooled CRISPR interference (CRISPRi) library was constructed in VL3834 (*S. pneumoniae* D39V $P_{lac}$-dCas9 Δ*mvaA-mvaS*). This CRISPRi library was grown in the presence of the dCas9 inducer, IPTG, and limiting amounts of mevalonic acid (100 μM). Cultures were sorted based on cell size (forward scatter [FSC]); 10% of the population with the highest and lowest values were sorted, as well as the centermost 70% of the population, which served as a control. sgRNAs from the sorted fractions were sequenced, and read counts were compared to identify gene depletions that led to changes in cell size. This plot shows the log₂ fold change of sgRNAs in different conditions and their statistical significance when the fraction with the highest FSC

*Figure 5 continued on next page*

*Figure 5 continued*

values (Max) was compared to the fraction of the population with the lowest FSC values (Min). **(J)** Quantitative analysis of microscopy images shows the changes in cell length upon single or double depletions of the first mevalonate operon (O1, *mvaS-mvaA*), DivIVA, or Pbp2x. Data are represented as violin plots with the mean ± SEM indicated, n≥3, with each repeat containing >90 cells except for the double O1 Pbp2x depletion where not enough surviving cells could be visualized and the threshold was put at 10 cells.

The online version of this article includes the following figure supplement(s) for figure 5:

**Figure supplement 1.** An *S. pneumoniae* Δ*cps* mutant that is unable to produce capsule does not display an elongated phenotype nor a growth defect.

**Figure supplement 2.** A sCRilecs-seq (<u>s</u>ubsets of <u>CR</u>ISPR <u>i</u>nterference <u>l</u>ibraries <u>e</u>xtracted by fluorescence activated <u>c</u>ell <u>s</u>orting coupled to next generation <u>seq</u>uencing) screen on mevalonate depleted cells to study the underlying genetic network.

(*Barendt et al., 2009*), demonstrating that a shortage of capsule synthesis does not contribute to the mevalonate depletion phenotype. Next, we focused on the possible contribution of both teichoic acid and peptidoglycan synthesis by depleting the enzymes responsible for coupling the final cytoplasmic precursors to Und-P before transport to the outside of the cell. For peptidoglycan, it is known that the MurNAc-pentapeptide moiety that is produced in the cytoplasm is coupled to Und-P by MraY (*Bouhss et al., 2008*). Based on sequence similarity, bioinformatic analysis indicated that the enzyme that does the same for teichoic acid precursors in *S. pneumoniae* is SPV_1620 (*Denapaite et al., 2012*). Deletion/complementation strains for these genes were created, and the effect of depletion of either MraY or SPV_1620 was assessed (strain VL3585 and VL3586, respectively). Surprisingly, depletion of neither MraY nor SPV_1620 produced cell morphologies that resembled mevalonate operon depletions at the time we usually assess them (*Figure 5G–H*, time point 4 hr). However, since at this time cells were severely malformed and many had already lysed, we decided to follow these depletions through time and record morphologies at earlier time points. *Figure 5G* shows the progression of these depletions in terms of average cell length, while *Figure 5H* shows representative morphologies. From this analysis it became clear that, although depletion of both MraY and SPV_1620 led to increased cell length, the phenotype obtained upon SPV_1620 depletion differed profoundly from the phenotypes found when depleting mevalonate operons. Depletion of MraY, on the other hand, was highly similar to the mevalonate phenotype at early time points. We thus conclude that a deficiency in peptidoglycan precursors at the site of cell wall synthesis is mainly responsible for the elongated phenotype that is observed when components of the mevalonate pathway are depleted. Since we previously showed that this elongated phenotype is caused by continued peptidoglycan synthesis in the absence of cell constriction, it appears as though a certain threshold level of peptidoglycan precursors is needed for cells to divide. Cell elongation, on the other hand, can proceed with a lower amount of peptidoglycan precursors. Different explanations for such a threshold effect can be thought of and are elaborated on in our discussion.

## A sCRilecs-seq screen on mevalonate depleted cells to study the underlying genetic network

To identify pathways and genes that are particularly sensitive to reduced mevalonate levels and to obtain clues on why a lowered concentration of extracellular peptidoglycan precursors would allow elongation but not division, we performed a second sCRilecs-seq screen based on cell size (FSC) in a genetic background in which the first mevalonate operon was deleted (VL3834: D39V P*_lac_*-*dcas9 lacI* Δ*mvaS-mvaA*). This way, we could look for gene depletions that aggravate the observed phenotype, i.e., make cells even longer, and depletions that compensate for the mevalonate-dependent cell elongation. During construction of this CRISPRi library and prior to selection, cultures were grown in the presence of a high concentration of mevalonic acid (1 mM) which ensures wt growth and normal cell size even though the first mevalonate operon is deleted (see *Figure 3H–I*). After construction of the pooled CRISPRi library, cultures were grown with limiting amounts of external mevalonic acid (100 μm) to trigger the elongated phenotype caused by mevalonate deficiency (*Figure 5—figure supplement 2A*). Simultaneously, dCas9 expression was induced by the addition of 1 mM IPTG for 3.5 hr. 10% of the population with the highest and lowest FSC values were sorted by FACS, and the centermost 70% of the population were collected to serve as a control. Comparing sgRNA read counts from the collected fractions led to the identification of many significantly enriched targets in the populations

with the highest and lowest FSC values. We therefore limited our analysis to hits that are enriched or depleted by a factor 2 (*Figure 5I*, *Figure 5—figure supplement 2B* and *Supplementary file 3*).

GO enrichment analysis identified cell division and closely related GO categories as significantly overrepresented in elongated cells (*Supplementary file 4*). This suggests that depleting genes involved in cell division (such as significant hits *ftsZ*, *divIB*, *divIC,* and *ftsW*) will lead to additional cell elongation upon mevalonate depletion. However, there were only very few gene depletions that further increase the size of these mevalonate-deprived cells, indicating that cell length cannot be increased much more. Of note, depletion of both the second mevalonate operon and *uppP*, hits that were found to be highly enriched in the large-cell fraction in our original screen, no longer demonstrated differential FSC values here, indicating that they function in the same cell elongation mechanism as the deleted first mevalonate operon as expected. On the other hand, sgRNAs that are enriched in the small fraction of the population are often involved in protein expression (RNA biosynthesis, translation, tRNA aminoacylation, etc.) or energy metabolism (glycolysis, ATP synthesis coupled proton transport, etc.). Halting or interfering with protein expression or energy production therefore appears to reduce cell elongation caused by mevalonate deficiency. The results of these high-level analyses are clear; directly interfering with cell division reinforces the division block imposed by mevalonate depletion and inhibiting protein expression or energy production, both of which are necessary for growth in general, prevents cell elongation.

Additionally, when looking at individual genes and operons found among the significantly enriched or depleted hits, some interesting patterns emerge. For example, the sgRNA targeting *divIVA* was strongly depleted from the fraction of the population with the largest cell sizes and also highly enriched in the fraction with the smallest cells, indicating that the depletion of DivIVA hampers cell elongation upon mevalonate depletion. DivIVA activity was previously shown to be necessary for cell elongation, and DivIVA is known to be phosphorylated by the serine/threonine kinase StkP that is thought to constitute a molecular switch that governs elongation and division (*Vollmer et al., 2019*; *Fleurie et al., 2014*; *Beilharz et al., 2012*; *Fleurie et al., 2012*; *Nováková et al., 2010*). Indeed, we confirmed that a double depletion of the first mevalonate operon and DivIVA leads to reduced cell lengths in comparison to the depletion of mevalonate alone (*Figure 5J*), indicating that also upon mevalonate depletion DivIVA activity assists cell elongation. However, cells depleted for both mevalonate and DivIVA are still larger than wt cells, indicating that a limited amount of elongation still occurs. Elongation enforced by mevalonate depletion is therefore not fully dependent on DivIVA activity. Additionally, CRISPRi depletion of the SEDS(shape, elongation, division and sporulation) proteins FtsW and RodA had opposite effects on cell size in our screen, with FtsW depletion leading to increased cell sizes while RodA depletion prevented cells from obtaining the largest cell sizes (*Supplementary file 3*). These proteins are thought to be the primary transglycosylases responsible for peptidoglycan polymerization during cell division and cell elongation, respectively (*Cho et al., 2016*; *Meeske et al., 2016*; *Taguchi et al., 2019*; *Tsui et al., 2016*; *Perez et al., 2019*). Moreover, CRISPRi depletion of Pbp2b, the transpeptidase that works in conjunction with RodA to perform peripheral peptidoglycan synthesis (*Vollmer et al., 2019*; *Berg et al., 2013*; *Tsui et al., 2016*), showed up in the short-cell fraction (*Supplementary file 3*). These results indicate that RodA together with Pbp2b is at least partly responsible for cell elongation that occurs upon mevalonate depletion. Because Pbp2x, the transpeptidase that is essential for septal peptidoglycan synthesis (*Tsui et al., 2014*; *Massidda et al., 2013*), is encoded in an operon together with *mraY*, its effect on mevalonate-dependent cell elongation could not be directly assessed in the sCRilecs-seq screen. Therefore, we constructed a clean *pbp2x* depletion strain. As shown in *Figure 5J*, depletion of Pbp2x indeed led to an increase in cell length, as reported before (*Tsui et al., 2014*), and this phenotype was augmented under low mevalonate levels. However, elongation upon mevalonate depletion was less pronounced when *pbp2x* was simultaneously repressed, hinting at a role for this septation-specific PBP in elongation.

## The mevalonate pathway as druggable target in *S. pneumoniae*

Our results so far demonstrate that depletion of the mevalonate pathway leads to cell elongation, and these morphological effects can be enhanced by targeting other cell division pathways. Since amoxicillin also causes elongation in *S. pneumoniae*, we set out to design a combination treatment strategy using amoxicillin and exploiting the mevalonate depletion phenotype characterized here. In a first step, we confirmed that the antibacterial effect of amoxicillin – but not that of the fluoroquinolone

ciprofloxacin – is increased upon mevalonate depletion, since amoxicillin could inhibit growth already at sub-MIC concentrations when mevalonate became limiting (*Figure 6—figure supplement 1A-B*). These results demonstrate that, in principle, it is possible to design a treatment strategy in which amoxicillin is potentiated by targeting the mevalonate pathway.

In a second step, we looked for chemical compounds that could block either the mevalonate pathway or the downstream production of Und-P. Three compounds were selected; simvastatin and farnesol are expected to block the HMG-CoA reductase MvaA that catalyzes one of the first steps of the mevalonate pathway (*Masadeh et al., 2012*; *Bergman et al., 2011*; *Kaneko et al., 2011*), while clomiphene was shown to inhibit UppS in *S. aureus* and is counteracted by the addition of exogenous Und-P[33]. To confirm the specificity of these compounds for the mevalonate pathway or downstream production of Und-P, we first tested whether they could induce the expected cell elongation in *S. pneumoniae*. Unfortunately, simvastatin and farnesol were not able to cause cell elongation. Farnesol, which was shown to be effective in *S. aureus*, most likely does not work in *S. pneumoniae*. Simvastatin on the other hand is a specific inhibitor of eukaryotic-type HMG-CoA reductase and is expected to be far less active against the bacterial counterpart of this enzyme (*Perez and Rodríguez-Concepción,*

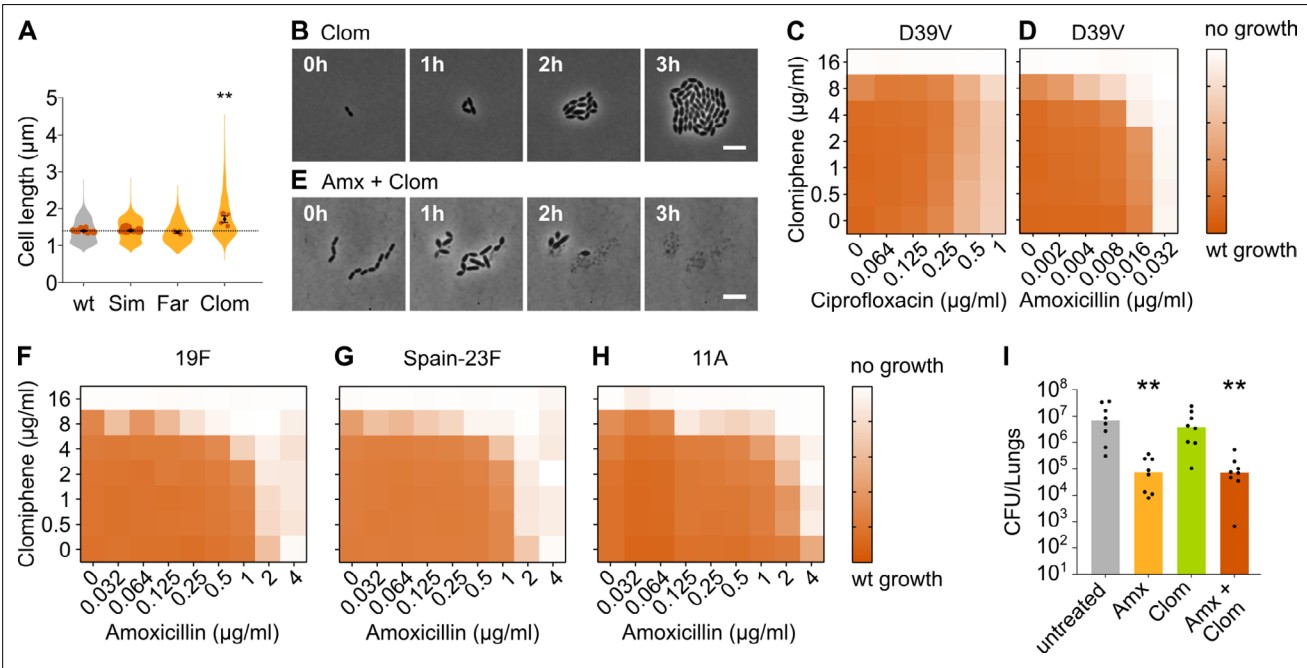

**Figure 6.** Clomiphene, an inhibitor of undecaprenyl phosphate (Und-P) production, potentiates amoxicillin. (**A**) The effect of several potential inhibitors of Und-P production on the cell length of *S. pneumoniae* D39V (VL333) was tested (Sim 4 µg/ml, Far 4 µg/ml, and Clom 8 µg/ml). Quantitative analysis of microscopy images shows that cell length increased upon treatment with clomiphene. Data are represented as violin plots with the mean cell length of every biological repeat indicated with orange dots. The size of these dots indicates the number of cells recorded in each repeat, ranging from 177 to 6464 cells. Black dots represent the mean ± SEM of all recorded means, n≥3. Two-sided Wilcoxon signed rank tests were performed against wt without treatment as control group (dotted line), and p values were adjusted with a false discovery rate (FDR) correction; ** p<0.01. (**B**) Snapshot images of phase contrast time-lapse microscopy of *S. pneumoniae* D39V (VL333) in the presence of clomiphene (8 µg/ml). Scale bar, 5 µm. (**C–D**) OD595nm growth curves were constructed for *S. pneumoniae* D39V (VL333) in the presence of different concentrations of clomiphene and ciprofloxacin (**C**) or amoxicillin (**D**). Heatmaps of the area under the resulting growth curves are shown. Number of biological repeats for all experiments, n≥3. (**E**) Snapshot images of phase contrast time-lapse microscopy of *S. pneumoniae* D39V (VL333) in the presence of clomiphene (8 µg/ml) and amoxicillin (0.016 µg/ml). Scale bar, 5 µm. (**F–H**) OD595nm growth curves were constructed for *S. pneumoniae* 19F (**F**), Spain-23F (**G**), and 11A (**H**) in the presence of different concentrations of clomiphene and amoxicillin. Heatmaps of the area under the resulting growth curves are shown. Number of biological repeats for all experiments, n≥3. (**I**) The effect of the combination treatment with amoxicillin and clomiphene was tested in vivo using a pneumonia superinfection model with a clinical isolate of *S. pneumoniae* serotype 19F. Mice (n=8 per group) were infected intranasally first with H3N2 virus and then 7 days later with pneumococcus 19F. Mice were treated at 8 hr and 12 hr with clomiphene, amoxicillin, combination of both, or left untreated. Lungs were collected 24 hr post-infection to measure the bacterial load. CFU counts for individual mice are shown, and the bars represent the median value. The data were compared in a Kruskall-Wallis test (one-way ANOVA), ** p<0.01. wt: wildtype; Sim: simvastatin; Far: farnesol; Clom: clomiphene; Amx: amoxicillin.

The online version of this article includes the following figure supplement(s) for figure 6:

**Figure supplement 1.** The negative effect of clomiphene, methicillin, and amoxicillin on growth is exacerbated by mevalonate depletion.

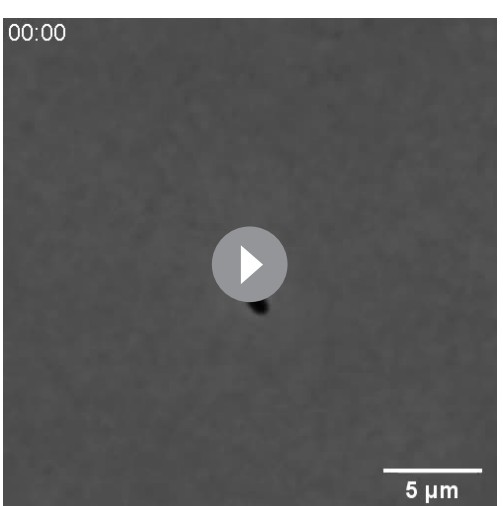

**Video 5.** An *S. pneumoniae* D39V (VL333) growing on agarose pads of C+Y medium supplemented with 8 µg/ml clomiphene.
https://elifesciences.org/articles/75607/figures#video5

**Table 1.** MIC values for amoxicillin/clomiphene and ciprofloxacin/clomiphene drug combinations (µg/ml).

| | Clomiphene | | |
| --- | --- | --- | --- |
| | 0 µg/ml | 4 µg/ml | 8 µg/ml |
| ***S. pneumoniae* D39V** | | | |
| Ciprofloxacin | 1 | 1 | 1 |
| Amoxicillin | 0.032 | 0.016 | 0.004 |
| ***S. pneumoniae* 19F** | | | |
| Amoxicillin | 2 | 1 | 0.125 |
| ***S. pneumoniae* Spain-23F** | | | |
| Amoxicillin | 2 | 1 | 0.032 |
| ***S. pneumoniae* 11A** | | | |
| Amoxicillin | 4 | 2 | 0.125 |

*2013*). However, the addition of clomiphene did result in filamentation under our conditions (*Figure 6A*). We therefore selected this compound for further testing. Indeed, time-lapse analysis showed considerable cell elongation in the presence of clomiphene, albeit less than upon depletion of the mevalonate operons potentially due to the lag in severe and lethal effects upon gradual genetic depletion (*Figure 6B* and *Video 5*). Additionally, clomiphene most likely targets the production of Und-P, since sub-MIC concentrations of clomiphene led to severe growth defects in cultures that were slightly depleted for mevalonate (*Figure 6—figure supplement 1C*).

We next combined clomiphene with either amoxicillin or ciprofloxacin at several different concentrations and monitored the growth of an *S. pneumoniae* D39V wt strain by measuring OD. The success of growth was quantified by calculating the area under the curve for a growth period of 10 hr. As can be seen in *Figure 6C*, clomiphene did not affect the efficacy of ciprofloxacin. However, for amoxicillin, potentiation by clomiphene was observed (*Figure 6D*). This is consistent with time-lapse analyses that show that survival of *S. pneumoniae* was much more affected when both compounds were combined (*Figure 6E* and *Video 6*). Indeed, in the presence of clomiphene, the concentration of amoxicillin necessary to block pneumococcal growth decreased by a factor 2–8 (*Table 1*).

## Resensitizing antibiotic-resistant *S. pneumoniae* strains using clomiphene

We next tested whether the potentiation by clomiphene is also present in clinically relevant amoxicillin-resistant *S. pneumoniae*. We therefore assessed the growth of a panel of resistant strains in the presence of different concentrations of amoxicillin and clomiphene. The strains we tested are clinical isolates of strains 19F, Spain-23F, and 11 A (*Càmara et al., 2018*; *Imöhl et al., 2010*). Results are presented in *Figure 6F–H* and *Table 1* and show that clomiphene potentiated the antimicrobial activity of amoxicillin also against clinical resistant strains. Moreover, the potentiation was considerably stronger in these resistant genetic backgrounds. Indeed, the concentration of amoxicillin necessary to block growth of these resistant strains decreased by a factor 16–64 in the presence of 8 µg/ml of clomiphene (*Table 1*), thereby reducing resistance below the EUCAST clinical breakpoint for sensitivity (0.5 µg/ml) (*EUCAST,*

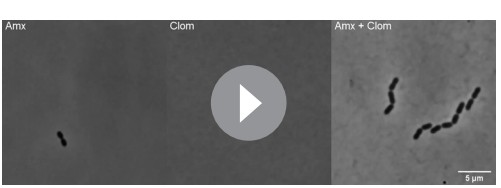

**Video 6.** An *S. pneumoniae* D39V (VL333) growing on agarose pads of C+Y medium supplemented with 0.016 µg/ml amoxicillin (left), 8 µg/ml clomiphene (middle), or both 0.016 µg/ml amoxicillin and 8 µg/ml clomiphene (right).
https://elifesciences.org/articles/75607/figures#video6

*2021*). Our rationally designed combination treatment strategy can thus resensitize amoxicillin-resistant *S. pneumoniae* strains so that they once again become fully susceptible to amoxicillin in vitro. This treatment strategy could therefore prove useful in the fight against amoxicillin-resistant *S. pneumoniae* infections.

We next tested whether this treatment strategy would prove useful in an in vivo setting, using a murine pneumonia disease model (see Materials and methods) (*Matarazzo et al., 2019*). Mice were infected on day 1 with influenza A virus and superinfected on day 7 with *S. pneumoniae* 19F. Animals were treated with 5 mg of clomiphene and/or 1 mg of amoxicillin at 8 and 12 hr post-pneumococcal infection, respectively. At 24 hr post infection, the bacterial load in the lung was determined. Whereas treatment with amoxicillin alone significantly lowered bacterial counts, administering only clomiphene did not influence the bacterial load (*Figure 6I*). Treating mice with both clomiphene and amoxicillin did not show a stronger effect than treatment with amoxicillin alone, meaning that the potentiation of amoxicillin we observed in vitro could not be detected in vivo (*Figure 6I*). The same was true for CFU counts in the spleen (*Figure 6—figure supplement 1D*). To figure out why our combination treatment strategy was not effective in vivo, we tried to estimate the active concentration of clomiphene in the lung. Our in vitro tests showed that potentiation occurs at concentrations of clomiphene of 4 μg/ml and higher, but these concentrations might not be achieved in vivo. We therefore tested the growth of *S. pneumoniae* D39V in vitro in the presence of bronchoalveolar lavage (BAL) fluid collected from mice treated with clomiphene. This experiment indicated that the active concentration of clomiphene in the lung was lower than 4 μg/ml, which could explain the absence of amoxicillin potentiation (*Figure 6—figure supplement 1E*).

## Discussion

Here, we rationally designed a combination treatment strategy that resensitizes resistant *S. pneumoniae* strains toward clinically relevant concentrations of amoxicillin, one of the most widely used antibiotics to fight this human opportunistic pathogen (*Jacobs, 2004*; *Bradley et al., 2011*; *Thiem et al., 2011*). By combining amoxicillin with the FDA-approved fertility drug clomiphene (*Dickey and Holtkamp, 1996*; *Huppert, 1979*; *Nasseri and Ledger, 2001*), we were able to reduce amoxicillin MIC values of resistant strains 16–64-fold in vitro, thereby decreasing them below the EUCAST clinical breakpoint for sensitivity.

### sCRilecs-seq identifies several targets that regulate *S. pneumoniae* cell size

This combination treatment strategy was based on the results of the sCRilecs-seq screen developed here. sCRilecs-seq is a high-throughput single-cell-based screening approach that relies on genome-wide CRISPRi depletion combined with sorting of cells that display a phenotype of interest. We here screened for CRISPRi gene depletions that, like amoxicillin treatment, result in cell elongation. The rationale behind this strategy is that synergy is most often detected between compounds that target the same process (*Brochado et al., 2018*). We show that hits identified by our sCRilecs-seq screen are reliable, accurate, and contain biologically relevant information. Interestingly, significant hits include a number of genes of unknown function that we can now implicate in the maintenance of proper cell size. Our sCRilecs-seq approach is thus able to uncover novel gene functions. On the other hand, several genes known to be involved in cell size regulation were not identified as significant hits, the most obvious example being the divisome-organizing cell division gene *ftsZ*. In the case of *ftsZ*, this false negative result might be caused by the core essential nature of this gene which would lead to low read counts for the corresponding sgRNA in the entire population. Since these low read counts are expected to strongly decrease statistical power, it is unsurprising that some essential genes are missed by our screen. However, such false negatives could be caused by a variety of other reasons as well. For example, simultaneous repression of all genes in an operon can obscure phenotypes expected upon repression of individual genes. Additionally, we observe a relatively high amount of variation between different repeats, which can also contribute to false negative results. Finally, depletion strains sensitive to mechanical vortexing could potentially be missed due to premature lysis, although no indications for such biased loss of strains were found. The latter issue could potentially be solved by working with

an unencapsulated mutant strain that does not form cell chains which would eliminate the need for vortexing and is expected to exacerbate cell division phenotypes (*Barendt et al., 2009*).

Although we used the sCRilecs-seq method to screen for increases in cell size, this approach could be adapted to assess any phenotype of interest that can be measured by flow cytometry. In contrast to 'classic' CRISPRi-seq screens (*Cui, 2018*; *Liu et al., 2021*; *de Wet et al., 2018*; *Lee et al., 2019*; *Rousset et al., 2018*; *Wang et al., 2018*), our approach is therefore not limited to measuring changes in fitness but can be used to evaluate a wide array of different phenotypes, of which changes in cell size is just one possibility. In fact, we believe that the sCRilecs-seq method could be further improved when selecting for phenotypes other than cell size. We expect variation in experimental results to decrease when selecting for phenotypes that are better separated and display a smaller spread than FSC values do. Additionally, assessing phenotypes that are not associated with essential genes would eliminate the problem of decreased statistical power for corresponding sgRNAs (see above).

## Depletion of mevalonate pathway genes leads to cell elongation through inhibition of cell division

The sCRilecs-seq screen performed here revealed an important role for the mevalonate pathway in maintaining proper cell size. Indeed, depletion of the genes involved in this pathway led to a very large increase in cell length. Results obtained strongly indicate that this cell elongation is due to a deficiency in the production of Und-P, the lipid carrier that translocates all different cell envelope precursors across the cell membrane (*Bouhss et al., 2008*), and subsequent limitation of the amount of peptidoglycan building blocks that is available for cell wall synthesis. This limiting amount of peptidoglycan precursors led to filamentation in *S. pneumoniae*. Interestingly, chemically blocking Und-P production in the rod-shaped bacterium *Bacillus subtilis* led to a vastly different phenotype that more resembles a deficiency in wall teichoic acids than impairment in peptidoglycan synthesis (*Farha et al., 2015*). These results indicate that the Und-P-dependent elongation phenotype is not universally conserved and might be specifically tied to the regulatory mechanisms responsible for morphogenesis of *S. pneumoniae* and possibly other related bacteria.

*S. pneumoniae* filamentation caused by depletion of mevalonate pathway components is linked to changes in peptidoglycan metabolism. Whereas the peptidoglycan synthesis rate in general decreased, cell elongation continued long after constriction for cell division was inhibited. This observation points toward a difference in affinity and/or regulation of elongation and constriction, where it appears that a certain threshold concentration of peptidoglycan precursors needs to be exceeded for cell division to take place. Several potential explanations for this observation can be put forward. First, it is possible that peptidoglycan synthesis enzymes dedicated to cell division have a lower affinity for peptidoglycan precursors than their counterparts that function in cell elongation. When cell wall precursors become limiting, the enzymes involved in septal ring closure would be outcompeted and cell division would cease while elongation still occurs. Alternatively, it is possible that depletion of mevalonate pathway components and subsequent peptidoglycan precursor shortage induces transcriptional or post-transcriptional regulatory pathways that differentially affect the activity of protein complexes involved in cell elongation and division, thereby leading to filamentation. On the other hand, it has previously been suggested that a more complex switch-like regulatory mechanism decides whether cells elongate or divide. The serine/threonine kinase StkP and its cognate phosphatase PhpP was suggested to constitute a molecular switch that coordinates septal and peripheral peptidoglycan synthesis through phosphorylation and dephosphorylation of its targets (*Vollmer et al., 2019*; *Fleurie et al., 2014*; *Beilharz et al., 2012*; *Fleurie et al., 2012*). This switch is regulated by GpsB, which is necessary for maximal phosphorylation in *S. pneumoniae* and is essential for viability (*Rued et al., 2017*; *Land et al., 2013*; *Cleverley et al., 2019*). Interestingly, depletion of GpsB leads to an elongated phenotype with unconstricted cell division sites (*Rued et al., 2017*; *Land et al., 2013*) that are highly reminiscent of the mevalonate depletion phenotype, hinting at possible crosstalk between both pathways. Results presented here indicate that, if such a switch exists, the decision between elongation and division would depend on the amount of peptidoglycan precursors that is exported. Since StkP contains PASTA domains that are known to bind peptidoglycan precursors (*Vollmer et al., 2019*; *Beilharz et al., 2012*; *Fleurie et al., 2012*; *Zucchini et al., 2018*), it has been suggested that its activity is indeed regulated in response to the concentration of peptidoglycan building blocks present (*Vollmer et al., 2019*; *Beilharz et al., 2012*; *Fleurie et al., 2012*). However, it was previously

shown that even though PASTA domains are necessary for StkP activation, peptidoglycan binding is not required (*Zucchini et al., 2018*), arguing against StkP being a sensor for peptidoglycan precursor levels. Moreover, several of our observations do not fit into this switch-like regulatory model for peptidoglycan synthesis. First, the depletion of DivIVA, which is thought to be an integral part of the switch between elongation and division (*Rued et al., 2017*; *Fleurie et al., 2014*; *Nováková et al., 2010*), still allowed a modest amount of elongation to occur upon mevalonate depletion. Second, depletion of the division-specific transpeptidase Pbp2x limits the amount of elongation seen during mevalonate depletion. Additionally, the suggested switch-like behavior between peripheral and septal peptidoglycan synthesis has recently been put into question (*Trouve et al., 2021*). Detailed microscopic analysis of newly synthesized, fluorescently labeled peptidoglycan indicated that peripheral and septal peptidoglycan synthesis overlap throughout large parts of the cell cycle. It was shown that septal cell wall synthesis and splitting of the newly formed septal peptidoglycan occur throughout the early- and mid-cell cycle stages. The split septal peptidoglycan then becomes part of the lateral wall where peripheral peptidoglycan synthesis occurs and is inserted (*Trouve et al., 2021*). These results imply that a septum can only be formed when the septal peptidoglycan synthesis rate is higher than the splitting rate (*Trouve et al., 2021*). Our results can be interpreted in light of this model; FDAA labeling of active peptidoglycan synthesis showed that reduced precursor concentrations decrease the general peptidoglycan synthesis rate. This might render septal synthesis too slow in comparison to peptidoglycan splitting so that constriction is inhibited and no septum can be formed, even though septal cell wall synthesis is active. The slow rate of ongoing septal and peripheral peptidoglycan synthesis would then result in continued cell elongation. This interpretation of our results fit with observations regarding DivIVA and Pbp2x in the following way. In the absence of DivIVA, peripheral peptidoglycan synthesis is halted, but cell elongation upon mevalonate depletion can still occur due to the slowly ongoing septal cell wall synthesis which, combined with fast peptidoglycan splitting, results in elongation. Following the same logic, depletion of the septal cell wall synthesis enzyme Pbp2x now limits mevalonate-dependent elongation because the splitting of the septal wall no longer contributes to increased cell lengths. If true, slowing down septal peptidoglycan splitting should aid in constriction and should lower cell lengths in conditions where mevalonate is depleted. Indeed, downregulation of the proposed cell wall splitting enzyme PcsB reduces cell size upon mevalonate depletion according to our second sCRilecs-seq screen results. Further research will be necessary to conclusively determine how septal and peripheral peptidoglycan synthesis are regulated and balanced in ovococci such as *S. pneumoniae* and how the peptidoglycan precursor concentration influences this mechanism.

## Inhibition of Und-P synthesis by clomiphene resensitizes resistant *S. pneumoniae* to amoxicillin

We successfully exploited the knowledge gained on the mevalonate depletion phenotype to fight amoxicillin-resistant *S. pneumoniae* strains. To do so, we confirmed that clomiphene, a compound known to block Und-P production and potentiate β-lactam antibiotics in *S. aureus* and *B. subtilis* (*Farha et al., 2015*), elicits the elongated mevalonate depletion phenotype in *S. pneumoniae* and therefore most likely also inhibits Und-P production in this bacterium. Additionally, MIC values for clomiphene were highly similar for all three bacterial species. The clomiphene MIC for *S. aureus* and *B. subtilis* was reported to be 8 µg/ml and is 16 µg/ml for MRSA (*Farha et al., 2015*). Our results show that the clomiphene MIC value for *S. pneumoniae* D39V and selected clinical isolates is 16 µg/ml. We next combined clomiphene with the β-lactam antibiotic amoxicillin that preferentially interferes with cell division (*Berg et al., 2013*; *Kocaoglu et al., 2015*). Results demonstrated that clomiphene can enhance the antimicrobial effect of amoxicillin in an amoxicillin-sensitive, virulent serotype 2 strain (D39V) approximately eightfold. Also for MRSA and *B. subtilis*, it has been shown that the antimicrobial activity of a variety of β-lactam antibiotics is increased in the presence of clomiphene. The sensitivity of MRSA is increased by a factor of approximately 8, while this increase in sensitivity was less pronounced for *B. subtilis* (*Farha et al., 2015*). Interestingly, potentiation by clomiphene becomes much stronger in amoxicillin-resistant *S. pneumoniae* strains such as 19F, Spain-23F, and 11A where the sensitivity toward amoxicillin is increased 16- to 64-fold. The potentiation by clomiphene is therefore proven to be more powerful in amoxicillin-resistant *S. pneumoniae* strains than in amoxicillin-sensitive *S. pneumoniae* as well as MRSA and *B. subtilis*. In *S. pneumoniae*, resistance toward β-lactam antibiotics is mostly caused by mosaic PBPs. These resistance-conferring mosaic PBPs are formed by

recombination events following horizontal gene transfer from β-lactam resistant donor strains (*Jensen et al., 2015*; *Hakenbeck et al., 2012*; *Gibson et al., 2021*). It has been shown that many resistance-conferring PBP mutations are associated with considerable fitness costs that need to be compensated for (*Albarracín Orio et al., 2011*; *Zerfass et al., 2009*; *Smith and Klugman, 2001*) and that mosaic PBPs significantly alter the cell wall structure, potentially by a decrease in affinity for their natural substrate (*Garcia-Bustos and Tomasz, 1990*). We therefore speculate that these mosaic PBPs display suboptimal activity which allows them to remain active during amoxicillin treatment but fail to carry out their task if the amount of peptidoglycan precursors available to them becomes limiting due to a deficiency in Und-P production. We believe that these suboptimal, resistance-conferring, mosaic PBPs are the reason why clomiphene has a much stronger potentiation effect in these strains than in all other tested strains and species.

Whereas this combination treatment strategy works remarkably well in vitro, in vivo results using a murine pneumonia disease model did not show benefits of combining amoxicillin treatment with clomiphene, likely because the in vivo concentration of active clomiphene was too low in the lungs. Nonetheless, we believe that clomiphene represents a promising starting point for the development of an optimized antibacterial compound that can be used in combination with amoxicillin and potentially other β-lactams. Clomiphene is an FDA-approved prodrug that is administered as a racemic mixture of two stereoisomers and is metabolized by the liver into active compounds that stimulate ovulation in anovulatory women (*Dickey and Holtkamp, 1996*; *Huppert, 1979*; *Nasseri and Ledger, 2001*; *Mürdter et al., 2012*). However, since we observe antibacterial effects and amoxicillin potentiation in vitro, it seems likely that one or both of the prodrug isomers exert the desired effect. Determining which clomiphene stereoisomer has the highest antibacterial activity and designing non-metabolizable derivatives that are active at lower concentrations would be the first step to further optimize the here proposed combination treatment strategy for in vivo use. We hope that such an optimized antibacterial compound can be exploited for the eradication of amoxicillin-resistant *S. pneumoniae* infections and could potentially also target other species and/or strains with a different resistance profile toward β-lactam antibiotics. We therefore believe that further investigation into this combination treatment strategy holds much promise in combating the ever-increasing amount of antibiotic-resistant bacterial infections.

## Materials and methods

### Key resources table

| Reagent type (species) or resource | Designation | Source or reference | Identifiers | Additional information |
|---|---|---|---|---|
| Strain and strain background (*S. pneumoniae*) | D39V | doi:10.1093/nar/gky725 | VL001 | Serotype 2 and wildtype |
| Strain and strain background (*S. pneumoniae*) | 19F | This paper | VL4303 | Serotype 19F and clinical isolate |
| Strain and strain background (*S. pneumoniae*) | Spain-23F | German National Reference Center for Streptococci | VL1306 | Serotype 23F, PMEN1, and clinical isolate |
| Strain and strain background (*S. pneumoniae*) | 11A | German National Reference Center for Streptococci, doi:10.1093/jac/dky305 | VL1313 | Serotype 11A, PMEN3, and clinical isolate |
| Strain and strain background (*Mus musculus*) | C57BL/6JRj, male, 8 weeks old | Janvier Laboratories, Saint Berthevin, France | | |
| Genetic reagent (*S. pneumoniae*) | VL333 | Veening lab collection | VL333 | D39V *prs1::lacI-tetR-Gm* |
| Genetic reagent (*S. pneumoniae*) | VL567 | doi:10.1038/s41467-017-00903-y | VL567 | D39V Δ*cps::Cm* |
| Genetic reagent (*S. pneumoniae*) | VL1630 | Veening lab collection | VL1630 | D39V *ftsZ-mCherry-Ery bgaA::P$_{zn}$-gfp-stkP* |
| Genetic reagent (*S. pneumoniae*) | VL1998 | doi:10.15252/msb.20167449 | VL1998 | D39V *prs1::F6-lacI-Gm bgaA::P$_{lac}$-dCas9-Tc* |
| Genetic reagent (*S. pneumoniae*) | VL2226 | doi:10.1128/JB.02221-14 | VL2226 | D39V *hlpA-gfp-Cm* |
| Genetic reagent (*S. pneumoniae*) | VL3117 | This paper | VL3117 | VL1998 *hlpA-gfp-Cm ftsZ-mCherry-Ery* |
| Genetic reagent (*S. pneumoniae*) | VL3404 | This paper | VL3404 | VL333 *hlpA-gfp-Cm ftsZ-mCherry-Ery* |
| Genetic reagent (*S. pneumoniae*) | VL3565 | This paper | VL3565 | VL3404 *zip::P$_{lac}$-mvk-mvaD-mvaK2-fni-Spec* Δ*mvk-mvaD-mvaK2-fni::Km* |

*Continued on next page*

*Continued*

| Reagent type (species) or resource | Designation | Source or reference | Identifiers | Additional information |
|---|---|---|---|---|
| Genetic reagent (*S. pneumoniae*) | VL3567 | This paper | VL3567 | VL3404 *zip*::P$_{lac}$-*mvaS-mvaA-Spec ΔmvaS-mvaA::Km* |
| Genetic reagent (*S. pneumoniae*) | VL3584 | This paper | VL3584 | VL3404 *zip*::P$_{lac}$-*uppS-Spec ΔuppS::Km* |
| Genetic reagent (*S. pneumoniae*) | VL3585 | This paper | VL3585 | VL3404 *zip*::P$_{lac}$-*mraY-Spec ΔmraY::Km* |
| Genetic reagent (*S. pneumoniae*) | VL3586 | This paper | VL3586 | VL3404 *zip*::P$_{lac}$-*SPV_1620-Trm ΔSPV_1620::Km* |
| Genetic reagent (*S. pneumoniae*) | VL3671 | This paper | VL3671 | VL3117 *zip*::P3-*sgRNA035* |
| Genetic reagent (*S. pneumoniae*) | VL3672 | This paper | VL3672 | VL3117 *zip*::P3-*sgRNA036* |
| Genetic reagent (*S. pneumoniae*) | VL3673 | This paper | VL3673 | VL3117 *zip*::P3-*sgRNA046* |
| Genetic reagent (*S. pneumoniae*) | VL3674 | This paper | VL3674 | VL3117 *zip*::P3-*sgRNA087* |
| Genetic reagent (*S. pneumoniae*) | VL3675 | This paper | VL3675 | VL3117 *zip*::P3-*sgRNA100* |
| Genetic reagent (*S. pneumoniae*) | VL3676 | This paper | VL3676 | VL3117 *zip*::P3-*sgRNA121* |
| Genetic reagent (*S. pneumoniae*) | VL3677 | This paper | VL3677 | VL3117 *zip*::P3-*sgRNA135* |
| Genetic reagent (*S. pneumoniae*) | VL3678 | This paper | VL3678 | VL3117 *zip*::P3-*sgRNA355* |
| Genetic reagent (*S. pneumoniae*) | VL3679 | This paper | VL3679 | VL3117 *zip*::P3-*sgRNA432* |
| Genetic reagent (*S. pneumoniae*) | VL3680 | This paper | VL3680 | VL3117 *zip*::P3-*sgRNA455* |
| Genetic reagent (*S. pneumoniae*) | VL3681 | This paper | VL3681 | VL3117 *zip*::P3-*sgRNA460* |
| Genetic reagent (*S. pneumoniae*) | VL3682 | This paper | VL3682 | VL3117 *zip*::P3-*sgRNA461* |
| Genetic reagent (*S. pneumoniae*) | VL3683 | This paper | VL3683 | VL3117 *zip*::P3-*sgRNA503* |
| Genetic reagent (*S. pneumoniae*) | VL3684 | This paper | VL3684 | VL3117 *zip*::P3-*sgRNA573* |
| Genetic reagent (*S. pneumoniae*) | VL3685 | This paper | VL3685 | VL3117 *zip*::P3-*sgRNA583* |
| Genetic reagent (*S. pneumoniae*) | VL3686 | This paper | VL3686 | VL3117 *zip*::P3-*sgRNA590* |
| Genetic reagent (*S. pneumoniae*) | VL3687 | This paper | VL3687 | VL3117 *zip*::P3-*sgRNA628* |
| Genetic reagent (*S. pneumoniae*) | VL3688 | This paper | VL3688 | VL3117 *zip*::P3-*sgRNA673* |
| Genetic reagent (*S. pneumoniae*) | VL3689 | This paper | VL3689 | VL3117 *zip*::P3-*sgRNA757* |
| Genetic reagent (*S. pneumoniae*) | VL3690 | This paper | VL3690 | VL3117 *zip*::P3-*sgRNA780* |
| Genetic reagent (*S. pneumoniae*) | VL3691 | This paper | VL3691 | VL3117 *zip*::P3-*sgRNA785* |
| Genetic reagent (*S. pneumoniae*) | VL3692 | This paper | VL3692 | VL3117 *zip*::P3-*sgRNA796* |
| Genetic reagent (*S. pneumoniae*) | VL3693 | This paper | VL3693 | VL3117 *zip*::P3-*sgRNA813* |
| Genetic reagent (*S. pneumoniae*) | VL3694 | This paper | VL3694 | VL3117 *zip*::P3-*sgRNA822* |
| Genetic reagent (*S. pneumoniae*) | VL3695 | This paper | VL3695 | VL3117 *zip*::P3-*sgRNA824* |
| Genetic reagent (*S. pneumoniae*) | VL3696 | This paper | VL3696 | VL3117 *zip*::P3-*sgRNA900* |
| Genetic reagent (*S. pneumoniae*) | VL3697 | This paper | VL3697 | VL3117 *zip*::P3-*sgRNA1019* |
| Genetic reagent (*S. pneumoniae*) | VL3699 | This paper | VL3699 | VL3117 *zip*::P3-*sgRNA1064* |
| Genetic reagent (*S. pneumoniae*) | VL3700 | This paper | VL3700 | VL3117 *zip*::P3-*sgRNA1236* |
| Genetic reagent (*S. pneumoniae*) | VL3701 | This paper | VL3701 | VL3117 *zip*::P3-*sgRNA1240* |
| Genetic reagent (*S. pneumoniae*) | VL3702 | This paper | VL3702 | VL333 *ΔmvaS-mvaA::Km* |
| Genetic reagent (*S. pneumoniae*) | VL3708 | This paper | VL3708 | VL333 *zip*::P$_{lac}$-*mvk-mvaD-mvaK2-fni-Spec Δmvk-mvaD-mvaK2-fni::Km* |
| Genetic reagent (*S. pneumoniae*) | VL3709 | This paper | VL3709 | VL333 *zip*::P$_{lac}$-*mvaS-mvaA-Spec ΔmvaS-mvaA::Km* |
| Genetic reagent (*S. pneumoniae*) | VL3710 | This paper | VL3710 | VL333 *zip*::P$_{lac}$-*uppS-Spec ΔuppS::Km* |

*Continued on next page*

*Continued*

| Reagent type (species) or resource | Designation | Source or reference | Identifiers | Additional information |
|---|---|---|---|---|
| Genetic reagent (*S. pneumoniae*) | VL3711 | This paper | VL3711 | VL333 *zip::P*lac*-mraY-Spec ΔmraY::Km* |
| Genetic reagent (*S. pneumoniae*) | VL3712 | This paper | VL3712 | VL333 *zip::P*lac*-SPV_1620-Trm ΔSPV_1620::Km* |
| Genetic reagent (*S. pneumoniae*) | VL3834 | This paper | VL3834 | VL1998 *ΔmvaS-mvaA::Km* |
| Genetic reagent (*S. pneumoniae*) | VL4273 | This paper | VL4273 | VL333 *bgaA::P*tet*-pbp2x-Tc Δpbp2x::Ery* |
| Genetic reagent (*S. pneumoniae*) | VL4274 | This paper | VL4274 | VL3709 *bgaA::P*tet*-pbp2x-Tc Δpbp2x::Ery* |
| Genetic reagent (*S. pneumoniae*) | LD0001 | This paper | LD0001 | VL333 *bgaA::P*tet*-divIVA-Tc ΔdivIVA::Cm* |
| Genetic reagent (*S. pneumoniae*) | LD0002 | This paper | LD0002 | VL3709 *bgaA::P*tet*-divIVA-Tc ΔdivIVA::Cm* |
| Sequence-based reagent | OVL47 | This paper | PCR primer | GATTGTAACCGATTCATCTG |
| Sequence-based reagent | OVL48 | This paper | PCR primer | GGAATGCTTGGTCAAATCTA |
| Sequence-based reagent | OVL898 | This paper | PCR primer | CCAACAAGCTTCACAAAATAAACCG |
| Sequence-based reagent | OVL901 | This paper | PCR primer | CTTATCCGTTGCACGCTGACTC |
| Sequence-based reagent | OVL1369 | This paper | PCR primer | GTCTTCTTTTTTACCTTTAGTAACTACTAATCCTGCAC |
| Sequence-based reagent | OVL2077 | This paper | PCR primer | ATTCCTTCTTAACGCCCCAAGTTC |
| Sequence-based reagent | OVL2181 | This paper | PCR primer | GCGTCACGTCTCAGCATTATTTTTCCTCCTTATTTAT |
| Sequence-based reagent | OVL2182 | This paper | PCR primer | GCGTCACGTCTCACGGATCCCTCCAGTAACTCGAGAA |
| Sequence-based reagent | OVL2933 | This paper | PCR primer | GATCGGTCTCGAGGAATTTTCATATGAACAAAAATATAAAATATTCTCAA |
| Sequence-based reagent | OVL2934 | This paper | PCR primer | GATCGGTCTCGTTATTTCCTCCCGTTAAATAATAGATAACTATTAAAAAT |
| Sequence-based reagent | OVL3493 | This paper | PCR primer | GCCAATAAATTGCTTCCTTGTTTT |
| Sequence-based reagent | OVL3496 | This paper | PCR primer | ATGACACGGATTTTAAGAATAATTCTTTC |
| Sequence-based reagent | OVL3649 | This paper | PCR primer | TGTGTGGCTCTTCGAGAACTCGAGAAAAAAAACCGCGCCC |
| Sequence-based reagent | OVL3650 | This paper | PCR primer | TGTGTGGCTCTTCGGTTTCATTATTTTTCCTCCTTATTTATTTAGATCTTAATTGTGAGC |
| Sequence-based reagent | OVL3671 | This paper | PCR primer | CTGGTAGCTCTTCCAACATGCTGAAATGGGAAGACTTGCCTG |
| Sequence-based reagent | OVL3672 | This paper | PCR primer | CTGGTAGCTCTTCCTCTTTATTTTAGTACCTCAAACACGGTT |
| Sequence-based reagent | OVL3677 | This paper | PCR primer | GTATAGTAAGCTGGCAGAGAATATC |
| Sequence-based reagent | OVL3680 | This paper | PCR primer | ATACTTTTTAGGGACAGGATCAC |
| Sequence-based reagent | OVL3958 | This paper | PCR primer | GCGTCACGTCTCAATGCTCGTCTAGTAAAAGGAAAAAATGACAAAAAAA |
| Sequence-based reagent | OVL3959 | This paper | PCR primer | GCGTCACGTCTCATCCGTTACGCCTTTTCATCTGATCATTTG |

*Continued on next page*

*Continued*

| Reagent type (species) or resource | Designation | Source or reference | Identifiers | Additional information |
|---|---|---|---|---|
| Sequence-based reagent | OVL3962 | This paper | PCR primer | GCGTCACGTCTCAATG CAGTATAGAACGATTTT TTACATGAATGATAAAACAG |
| Sequence-based reagent | OVL3963 | This paper | PCR primer | GCGTCACGTCTCATCC GTTATGATCTTAAATTT TCGAGATAGCGCT |
| Sequence-based reagent | OVL3981 | This paper | PCR primer | GCGTCACGTCTCAAT GGCTAAAATGAGAA TATCACCGG |
| Sequence-based reagent | OVL3982 | This paper | PCR primer | GCGTCACGTCTCACTA AAACAATTCATCCAGTA AAATATAATATTTTATTTTCTCC |
| Sequence-based reagent | OVL3983 | This paper | PCR primer | GCGTCACGTCTCAGA GGACGCGCAAGCTG |
| Sequence-based reagent | OVL4061 | This paper | PCR primer | CACTACCAATTGG TGAAGTTGCT |
| Sequence-based reagent | OVL4062 | This paper | PCR primer | GCGTCACGTCTCACCT CTTTTTCCTTTTACTA GACGAAAAAACGTC |
| Sequence-based reagent | OVL4063 | This paper | PCR primer | GCGTCACGTCTCAT TAGGGCGTAACCAGCGCC |
| Sequence-based reagent | OVL4064 | This paper | PCR primer | TACAGGTACGAT GATTTTGGTCGT |
| Sequence-based reagent | OVL4069 | This paper | PCR primer | AGCTGAAGATAAA GCCTGTAACCA |
| Sequence-based reagent | OVL4070 | This paper | PCR primer | GCGTCACGTCTCAC CATGTAAAAAATCGT TCTATACTATTTT ATCACAAATGG |
| Sequence-based reagent | OVL4071 | This paper | PCR primer | GCGTCACGTCTCATT AGCATAAAAACTCA GACGAATCGGTCT |
| Sequence-based reagent | OVL4072 | This paper | PCR primer | ACAGCGCCGATTATTTCCTTTG |
| Sequence-based reagent | OVL4341 | This paper | PCR primer | GCGTCACGTCTCAAT TTATTTAGATCTTAA TTGTGAGCGCTC |
| Sequence-based reagent | OVL4583 | This paper | PCR primer | GCGTCACGTCTCAAAA TTTTTGAATAGGAATAA GATCATGTTTGGATTTT |
| Sequence-based reagent | OVL4584 | This paper | PCR primer | GCGTCACGTCTCATCC GCTAAACTCCTCCA AATCGGCG |
| Sequence-based reagent | OVL4585 | This paper | PCR primer | TCCAGATTTTTCTTAT GAGGAAACCTTATT |
| Sequence-based reagent | OVL4586 | This paper | PCR primer | GCGTCACGTCTCAC CATGATCTTATTCCT ATTCAAAAATCTA TCGTTTCATT |
| Sequence-based reagent | OVL4587 | This paper | PCR primer | GCGTCACGTCTCATT AGGGAGGAGTTTAG GAGGAAATATGACC |
| Sequence-based reagent | OVL4588 | This paper | PCR primer | CTGTACTGTCAACT ATCATAAAGATAATGGT |
| Sequence-based reagent | OVL4595 | This paper | PCR primer | GCGTCACGTCTCAAAA ATATTAACTTTAGGAG ACTAATATGTTTATTTCCATCAG |
| Sequence-based reagent | OVL4596 | This paper | PCR primer | GCGTCACGTCTCATC CGTTACATCAAATAC AAAATTGCGAGGGT |
| Sequence-based reagent | OVL4597 | This paper | PCR primer | AGATTGCTGACGA GAAAAATGGTG |

*Continued on next page*

*Continued*

| Reagent type (species) or resource | Designation | Source or reference | Identifiers | Additional information |
|---|---|---|---|---|
| Sequence-based reagent | OVL4598 | This paper | PCR primer | GCGTCACGTCTCACC ATATTAGTCTCCTAAA GTTAATGTAATTTT TTTAATGTCC |
| Sequence-based reagent | OVL4599 | This paper | PCR primer | GCGTCACGTCTCATT AGGAATGGCACCC TGATGTTTCA |
| Sequence-based reagent | OVL4600 | This paper | PCR primer | AATAAATCATCCATG TTGTTAAAATTATT AAAATTGTTGT |
| Sequence-based reagent | OVL4601 | This paper | PCR primer | GCGTCACGTCTCAC CATGCTGTTCTCC TTTGTTTTTATTATAC |
| Sequence-based reagent | OVL4602 | This paper | PCR primer | GCGTCACGTCTCATT AGAGTAGTCATAAG AAAATGAGTACAG |
| Sequence-based reagent | OVL5705 | This paper | PCR primer | GCGTCAGGTCTCAAT TTATTTAGATCTACTCT ATCAATGATAGAG TTATTATACTCT |
| Sequence-based reagent | OVL5706 | This paper | PCR primer | GCGTCAGGTCTCAGC GTAAGGAAATCCATTA TGTACTATTTCTG |
| Sequence-based reagent | OVL5707 | This paper | PCR primer | GCGTCAGGTCTCAAAA TTTAAGTAAGTGAGGA ATAGAATGCCAATTACA |
| Sequence-based reagent | OVL5708 | This paper | PCR primer | GCGTCAGGTCTCAAC GCCTACTTCTGGTTC TTCATACATTGGG |
| Sequence-based reagent | OVL5717 | This paper | PCR primer | GCGTCACGTCTCAAT TTATTTAGATCTACTCT ATCAATGATAGAG TTATTATACTCT |
| Sequence-based reagent | OVL5718 | This paper | PCR primer | GCGTCACGTCTCAGC GTAAGGAAATCCATT ATGTACTATTTCTG |
| Sequence-based reagent | OVL5727 | This paper | PCR primer | GCGTCAGGTCTCAAT GAACTTTAATAAAATT GATTTAGACAATTGGAAGAG |
| Sequence-based reagent | OVL5728 | This paper | PCR primer | GCGTCAGGTCTCAT TATAAAAGCCAGTC ATTAGGCCTATCT |
| Sequence-based reagent | OVL5729 | This paper | PCR primer | CTCCTTTTTTAACTCC TTTTATCAATCCTCA |
| Sequence-based reagent | OVL5730 | This paper | PCR primer | GCGTCAGGTCTCATCAT TCTATTCCTCACTTACT TAATAATAACTGGACG |
| Sequence-based reagent | OVL5731 | This paper | PCR primer | GCGTCAGGTCTCAATAA CTCCAGTGCATCCGACAGG |
| Sequence-based reagent | OVL5732 | This paper | PCR primer | ACCAAGTCCATTTCTTTACGTTTGAC |
| Sequence-based reagent | OVL6214 | This paper | PCR primer | GCGCGTAAGATTGAGCAA |
| Sequence-based reagent | OVL6215 | This paper | PCR primer | GATCGGTCTCATCCT ATCTTACTCCGCT ATTCTAATATTTTCA |
| Sequence-based reagent | OVL6216 | This paper | PCR primer | GATCGGTCTCGATAAAT CAAGGACATTAAAAA AATTACATTAACTT |
| Sequence-based reagent | OVL6217 | This paper | PCR primer | ACATCACCCATAAAGACCTTG |
| Sequence-based reagent | OVL6276 | This paper | PCR primer | GCGTCACGTCTCAAAA TTTAGAATAGCGGAG TAAGATATGAAGTGG |

*Continued*

| Reagent type (species) or resource | Designation | Source or reference | Identifiers | Additional information |
|---|---|---|---|---|
| Sequence-based reagent | OVL6277 | This paper | PCR primer | GCGTCACGTCTCAAC GCTTAGTCTCCTAAAGT TAATGTAATTTTTTTAATGTCC |
| Chemical compound and drug | Mevalonic acid lithium salt | Bio-Connect BV (BIPP) | HY-113071A | |
| Chemical compound and drug | Clomiphene citrate salt | Sigma - Aldrich | C6272 | |
| Chemical compound and drug | Clomid | Sanofi-Aventis | | Clomiphene citrate tablets USP, 50 mg tablets, used for in vivo assays |
| Chemical compound and drug | Amoxicillin | GlaxoSmithKline | | Clamoxyl for injection, used for in vivo assays |
| Chemical compound and drug | sBADA | Tocris Bioscience | 6659 | Work concentration 250 µM |
| Chemical compound and drug | RADA | Tocris Bioscience | 6649 | Work concentration 250 µM RRID:SCR_014237 |
| Software and algorithm | 2FAST2Q | doi:10.1101/2021.12.17.473121 | | https://github.com/veeninglab/2FAST2Q; *Veening Lab, 2022a* |
| Software and algorithm | DESeq2 | doi:10.1186/s13059-014-0550-8 | | https://bioconductor.org/ packages/release/bioc/ html/DESeq2.html |
| Software and algorithm | Fiji (Fiji Is Just ImageJ) | doi:10.1038/nmeth.2019 | | https://imagej.net/ software/fiji/downloads |
| Software and algorithm | Huygens software | Scientific Volume Imaging | | https://svi.nl/Huygens-Software |
| Software and algorithm | MicrobeJ | doi:10.1038/nmicrobiol.2016.77 | | https://www.microbej.com/ |
| Software and algorithm | BactMAP | doi:10.1111/mmi.14417 | | https://github.com/veeninglab/BactMAP; *Veening Lab, 2022b* |

## Bacterial strains and growth conditions

All pneumococcal strains used in this work are listed in the key resources table. Unless specified otherwise, strains used throughout this work are derivatives of *S. pneumoniae* D39V (*Slager et al., 2018*). In general, genomic changes were introduced by homologous recombination after transformation of a linear DNA molecule into *S. pneumoniae*. This linear DNA was either obtained through a one-step PCR reaction starting from a different strain that already carried the desired transformation product (e.g. *hlpA-gfp* and *ftsZ-mCherry*) or through golden gate cloning in which three different PCR products were ligated. The first and last of these PCR products were homologous to the up- and downstream regions of the genome where a deletion or insertion had to be made. In case of insertion of expression cassettes into the *zip* locus, these fragments also contained a P*lac* promoter and a specti-nomycin (or trimethoprim) resistance marker through amplification of fragments of the pPEPZ plasmid (*Keller et al., 2019*). In case of insertion of expression cassettes into the *bgaA* locus, these fragments also contained a P*tet* promoter and a tetracycline resistance marker through amplification of fragments from strain VL2212 (*Liu et al., 2021*). The third, middle fragment contained an antibiotic marker or a sequence to be inserted into the genome. Golden gate cloning sites were introduced into the PCR fragments as part of the primers. PCR fragments were digested with either BsaI, Esp3I, or SapI, followed by ligation and transformation. In case individual sgRNAs were cloned into *S. pneumoniae*, the sgRNA sequences were first inserted into the pPEPZ-sgRNAclone plasmid in *E. coli*, as described previously (*Liu et al., 2021*). These integrative plasmids were then transformed into *S. pneumoniae*. Transformation was performed with cells that were made competent by addition of the competence stimulating peptide (CSP-1) using a previously published protocol (*Synefiaridou and Veening, 2021*). Primers that were used for cloning are listed in the key resources table. *hlpA-gfp* was amplified from VL2226 with primers OVL47&48. *ftsZ-mCherry* was amplified from VL1630 with primers OVL898&901. Both PCR products were transformed to VL1998 to create VL3117. Complementation constructs for mevalonate operon 1 (*mvaS-mvaA*) and operon 2 (*mvk-mvaD-mvaK2-fni*) were made by amplification of the corresponding operons using primer pairs OVL3962&63 and OVL3958&59, respectively. Up- and downstream regions for insertion into the *zip* locus were amplified using primer pairs OVL3493&2,181 and OVL2182&3,496. For the complementation constructs of the *uppS* and *mraY* genes that were amplified using primers OVL4583&84 and OVL4595&96, respectively, up- and downstream regions of

the *zip* locus were created using OVL3493&4,341 and 2182&3,496. The complementation construct for SPV_1620 was created by amplification of the gene using OVL3671&72, and up- and downstream *zip* regions were produced using OVL3493&3,650 and 3649&3,496. Deletion of the corresponding native genes was performed by replacing them with a promoter-less kanamycin resistance cassette that was expressed using the native promoter of the deleted genes. The kanamycin cassette was amplified from the pPEPY plasmid (*Keller et al., 2019*) with primers OVL3981&82. The up- and downstream regions of the genes to be deleted were amplified using OVL4069&70 and OVL4071&72 for mevalonate operon 1 (*mvaS-mvaA*), OVL4585&86 and OVL4587&88 for *uppS*, OVL4597&98 and OVL4599&4,600 for *mraY,* and OVL3677&4,601 and OVL4602&3,680 for *SPV_1620*. Due to low expression levels (*Aprianto et al., 2018*), mevalonate operon 2 (*mvk-mvaD-mvaK2-fni*) was replaced with a Km cassette that carried its own constitutive promoter. This fragment was amplified with OVL3983&82 from pPEPY (*Keller et al., 2019*), and up- and downstream regions for this operon were amplified with primer pairs OVL4061&62 and OVL4063&64. A complementation construct for *divIVA* was made by amplification of this gene using primers OVL5707&08. Up- and downstream regions for insertion into the *bgaA* locus were amplified using primer pairs OVL2077&OVL5705 and OVL5706&1,369. Deletion of *divIVA* was performed by replacement with a promoter-less chloramphenicol resistance cassette that was expressed using the *divIVA* promoter. The chloramphenicol cassette was amplified from VL3117 to primers OVL5727&28. The up- and downstream regions of *divIVA* were amplified using OV5729&30 and OVL5731&32, respectively. A complementation construct for *pbp2x* was made by amplification of this gene using primers OVL6276&77. Up- and downstream regions for insertion into the *bgaA* locus were amplified using primer pairs OVL2077&OVL5717 and OVL5718&1,369. Deletion of *pbp2x* was performed by replacement with a promoter-less erythromycin resistance cassette that was expressed using the *pbp2x* promoter. The cassette was amplified from pJWV502 (*Liu et al., 2021*) to primers OVL2933&34. The up- and downstream regions of *pbp2x* were amplified using OV6214&15 and OVL6216&17, respectively.

Strains were grown in liquid C+Y medium (pH = 6.8) (*Domenech et al., 2018*) at 37°C without shaking under normal atmospheric conditions. Plating was performed inside Columbia agar with 3% sheep blood, and plates were incubated at 37°C in a controlled atmosphere containing 5% $CO_2$.

Antibiotics for selection were used at the following concentrations: chloramphenicol (Cm) 3 μg/ml, erythromycin (Ery) 0.5 μg/ml, gentamicin (Gm) 40 μg/ml, kanamycin (Km) 150 μg/ml, spectinomycin (Spec) 100 μg/ml, tetracycline (Tc) 0.5 μg/ml, and trimethoprim (Trm) 10 μg/ml. When necessary, $P_{lac}$ promoters were induced with various amounts of IPTG; 1 mM IPTG for induction of dCas9 for the sCRilecs-seq screens, 20 μM IPTG for complementation of mevalonate operon 1 (*mvaS*, *mvaZ*), 100 μM for complementation of *uppS,* and 1 mM IPTG for complementation of mevalonate operon 2 (*mvk*, *mvaD*, *mvaK2*, and *fni*), *mraY*, or *SPV_1620*. When necessary, $P_{tet}$ promoters were induced with 500 ng/ml anhydrotetracycline (aTc). Where appropriate, mevalonic acid was added to cultures at the concentration indicated in the text (ranging from 100 μM to 1 mM). When strains lacking the first mevalonate operon had to be grown under non-limiting conditions, 1 mM mevalonic acid was used. The concentrations of antibiotics and other compounds that were used to assess their antibacterial activity are indicated in the main text and/or figures and/or figure legends.

## Mechanical disruption of *S. pneumoniae* cell chains

Cultures were diluted 3× in PBS to a final volume of 1 ml in screw cap tubes. Tubes were placed into a FastPrep-24 5 G Instrument (MP Biomedicals) with QuickPrep-3 adapter and shaken with a speed of 6.0 m/s for 30 s. This protocol was validated using VL3117 that was grown for 3.5 hr in C+Y medium. Samples were either subjected to mechanical chain disruption using the FastPrep-24 5 G or not, after which both samples were analyzed by microscopy and flow cytometry.

## Construction of the CRISPRi libraries

CRISPRi libraries were constructed by transformation of the desired strains with a pool of 1499 different pPEPZ integrative plasmids carrying constitutively expressed sgRNA sequences that together target the entire genome (*Liu et al., 2021*). sgRNAs are under control of the constitutive P3 promoter, while the *dcas9* gene is inserted chromosomally under control of the inducible $P_{lac}$ promoter. The 1499 sgRNAs were designed to each target a specific operon. All sgRNA sequences together with their targets and potential off-targets were published previously (*Liu et al., 2021*). For the initial screen,

D39V P$_{lac}$-*dcas9 hlpA-gfp ftsZ-mCherry* (VL3117) was transformed. Our second sCRilecs-seq screen was performed after transformation of D39V P$_{lac}$-*dcas9 ΔmvaS ΔmvaA* (VL3834) in the presence of 1 mM mevalonic acid. For both libraries, at least $10^5$ individual transformants were obtained and collected.

## sCRilecs-seq screen

The transformed D39V P$_{lac}$-*dcas9 hlpA-gfp ftsZ-mCherry* (VL3117) library was grown for 3.5 hr in C+Y medium supplemented with 1 mM IPTG. The D39V P$_{lac}$-*dcas9 ΔmvaS ΔmvaA* (VL3834) library was grown for 3.5 hr in C+Y medium supplemented with 1 mM IPTG and 100 µM mevalonic acid. Cell chains were mechanically disrupted (see 'Mechanical disruption of *S. pneumoniae* cell chains') and further diluted into PBS 3–10× based on culture density.

Cells were sorted using a FACSAria IIIu Cell Sorter (BD Biosciences) equipped with violet, blue, and red lasers and a 70 µm nozzle. In case of the D39V P$_{lac}$-*dcas9 hlpA-gfp ftsZ-mCherry* (VL3117) library, cells were gated based on FSC, SSC, and GFP fluorescence. For the D39V P$_{lac}$-*dcas9 ΔmvaS ΔmvaA* (VL3834) library, cells were gated based on FSC and SSC only. For both libraries, cells with the 10% highest and lowest FSC values were collected, as well as 70% of the population located around the median FSC value. Flow rates and dilutions were adjusted to keep the efficiency of sorting as high as possible (and certainly above 85%) while not exceeding a sorting time of 60 min per sample. For every fraction, $1.5 \times 10^6$ cells were collected, and six different biological repeats were performed.

Cells were collected into 2 ml tubes, centrifuged at 18,000 g for 5 min, and pellets were stored at –20°C. Cells were lysed by dissolving pellets in 10 µl H$_2$O+0.025% DOC+0.05% SDS and incubating 20 min at 37°C, followed by 5 min incubation at 80°C. After samples were allowed to cool off, a colony PCR was performed using primers that contain index and adapter sequences necessary for Illumina sequencing. 10 µl of the lysed cell mixture was added to the PCR reaction as input DNA and 30 PCR cycles were performed. The amplicons were purified from a 2% agarose gel and Illumina sequenced on a MiniSeq according to manufacturer's instructions. Sequencing was performed with a custom sequencing protocol (*Liu et al., 2021*).

Data analysis was performed as described previously (*de Bakker et al., 2022*). The sgRNA sequences were recovered from the resulting reads, mapped onto the *S. pneumoniae* D39V sgRNA library, and counted using 2FAST2Q (*Bravo et al., 2021*), and the DESeq2 R package was used to define enrichments and associated p values for every sgRNA (*Love et al., 2014*). Briefly, DESeq2 takes raw count data as input and fits a binomial generalized linear model (GLM) to them. It normalizes by size factors, which account for both library size and library composition bias. Next, it tests for differential enrichment of each sgRNA between given conditions by testing the hypothesis that the log$_2$FC (the fitted sgRNA-wise GLM coefficients) significantly differs from a user-specified threshold. Here, we tested for any differential enrichment (absolute log$_2$FC >0) at a significance level of 0.05 (alpha = 0.05). Generated sgRNA-wise p values were adjusted for testing multiple comparisons with the Benjamini & Hochberg correction, controlling the false discovery rate (FDR). We added the sample information to the design formula of DESeq2 to account for differences between samples (paired data, as the Min, Middle, and Max fractions were taken from the same samples) (*Love et al., 2014*). For the initial sCRilecs-seq screen, we defined significant hits as sgRNAs from the fraction with the highest FSC values with an adjusted p value<0.1 and log$_2$FC >1 compared to counts from the corresponding control population. For the second sCRilecs-seq screen performed with a *ΔmvaS-mvaA* mutant limits was set at an adjusted p value<0.05 and log$_2$FC >1.

## Flow cytometry

Flow cytometry experiments were performed using NovoCyte 2,100YB (ACEA Biosciences) flow cytometer equipped with violet, blue, and red lasers. To validate sCRilecs-seq hits, strains were grown for 3.5 hr in C+Y medium with or without 1 mM IPTG. Cultures were diluted in PBS, and cell chains were disrupted as described above. Next, cells were incubated for 30 min at room temperature to mimic the average waiting time in the 60 min sorting step before FSC was measured. Cell gatings were chosen based on FSC, SSC, and GFP values, as was done during sorting.

## GO enrichment analyses

GO enrichment analyses were performed using the online GO Resource platform that is coupled to the PANTHER classification system analysis tool (*Ashburner et al., 2000*; *Mi et al., 2019*; *Consortium, 2019*). Validated significant sgRNA hits were translated into the spr identifiers that correspond to the targeted genes, which were used as the input gene set that was compared to the *S. pneumoniae* reference list provided by the platform. A PANTHER overrepresentation analysis was performed to identify biological processes that are overrepresented as defined by a Fisher's exact test using FDR-corrected p values. The significance cut-off was set at adjusted p value<0.05.

## OD growth curves

Cultures were grown until $OD_{595nm}$ 0.1 in C+Y medium under non-limiting conditions (deletion/complementation strains were grown with the appropriate amount of IPTG, aTc, or mevalonic acid, as listed in 'Bacterial strains and growth conditions'). Cultures were diluted 100× into C+Y medium supplemented or not with IPTG, aTc, or mevalonic acid and with or without the addition of antibacterial compounds, as indicated in the text. 300 µl of cell suspension was transferred into 96-well plates. When growth in the presence of BAL fluid was tested, cultures were diluted 100× into C+Y medium supplemented with ¼ BAL fluid. 200 µl of cell suspension was transferred into 96-well plates. In both cases, addition of compounds such as amoxicillin or clomiphene occurred at this stage by diluting stock concentrations into the wells of the 96-well plates (dilutions were chosen to always keep the DMSO(Dimethylsulfoxide) concentrations in the wells ≤1%). Growth was monitored by measuring $OD_{595\ nm}$ every 10 min using a TECAN Infinite F200 Pro. $OD_{595nm}$ values were normalized so that the lowest value measured during the first hour of growth was 0.001, the initial $OD_{595nm}$ value of the inoculum. In case the area under the curve needed to be calculated, values were log-transformed before this parameter was determined using GraphPad Prism 9.

## MIC measurements

MIC values were determined by constructing growth curves in C+Y medium (see above). The MIC was taken to be the concentration of the tested compound where the maximum $OD_{595nm}$ value obtained was less than 10% of the maximal $OD_{595nm}$ value obtained in the absence of the compound.

## Phase contrast and fluorescence microscopy

All microscopy images have been deposited on the EMBL-EBI BioImages Archive, with accession number S-BIAD477. For the determination of cell morphology and FtsZ localization, cultures were grown until $OD_{595\ nm}$ 0.1 in C+Y medium under non-limiting conditions (deletion/complementation strains were grown with the appropriate amount of IPTG or mevalonic acid, as listed in 'Bacterial strains and growth conditions'). Cultures were then diluted 100× into C+Y medium supplemented or not with IPTG or mevalonic acid and grown until the $OD_{595\ nm}$ of the wt strain reached 0.2. At this point, cultures were diluted to $OD_{595\ nm}$ of 0.1 and incubated for 45 min at 30°C in order to slow down growth prior to imaging. 1 ml of cell suspension was spun down (10,000 g for 2 min), and pellets were dissolved in 40 µl PBS. Cells were kept on ice prior to imaging.

To investigate peptidoglycan production using FDAAs, cultures were grown until $OD_{595\ nm}$ 0.1 in C+Y medium under non-limiting conditions (deletion/complementation strains were grown with the appropriate amount of IPTG, as listed in 'Bacterial strains and growth conditions'). Cultures were then diluted 100× into C+Y medium without IPTG and grown until the $OD_{595\ nm}$ of the wt strain reached 0.1. At this point, 250 µM sBADA was added, and cultures were incubated for 15 min at 37°C. Cells were washed three times with cold PBS (centrifuge at 10,000 g for 1 min at 4°C), and pellets were dissolved in C+Y medium containing 250 µM RADA. After incubating cells for 15 min at 37°C, the three wash steps were repeated. Pellets were dissolved in PBS and kept on ice prior to imaging.

Cells were imaged by placing 0.4 µl of cell suspension on pads made of PBS containing 1% agarose. Imaging was performed using a Leica DMi8 microscope with a sCMOS DFC9000 (Leica) camera and a SOLA light engine (Lumencor). Phase contrast images were acquired using transmission light with 100 ms exposure time. Leica DMi8 filter sets were used as follows: mCherry (Chroma 49017, Ex: 560/40 nm, BS: LP 590 nm, and Em: LP 590 nm) with exposure time 700 ms, sBADA (Ex: 470/40 nm Chroma ET470/40×, BS: LP 498 Leica 11536022, and Em: 520/40 nm Chroma ET520/40 m) with exposure time 200 ms, and RADA (Chroma 49017, Ex: 560/40 nm, BS: LP 590 nm, and Em: LP 590 nm)

with exposure time 100 ms. Images were processed using ImageJ, and deconvolution was performed using Huygens software (Scientific Volume Imaging) with standard settings using 15 iterations for FtsZ-mCherry and sBADA and 25 iterations for RADA. Quantification of cell length and other properties was done using MicrobeJ (*Ducret et al., 2016*) and BactMAP (*van Raaphorst et al., 2020*).

For the analysis and statistical comparison of cell size across different mutants and conditions (*Figure 1C*; *Figure 2D*, *Figure 3D, F and H*; *Figure 3—figure supplement 1A-C*; *Figure 5J*; *Figure 6A*), the following approach was used (*Lord et al., 2020*). For every mutant and/or condition, cell size was recorded for at least three biologically independent repeats. The one exception is the depletion of *mraY* at time point 4.5 hr where only two independent repeats could be analyzed. At this late time point, the culture is most often taken over by suppressor mutants with normal morphology. Despite many attempts, we failed to obtain a third repeat in which the *mraY* depletion phenotype was still apparent. For every repeat, at least 100 cells were recorded unless mentioned otherwise, and their average cell length was determined. These average cell lengths were used to calculate the mean and SEM values that are shown in the main figures. The average cell lengths of each repeat were also used to determine any statistically significant differences using a Wilcoxon test with FDR corrected p values.

## Transmission electron microscopy

Cultures were grown until $OD_{595\,nm}$ 0.1 in C+Y medium under non-limiting conditions (deletion/complementation strains were grown with the appropriate amount of IPTG, as listed in 'Bacterial strains and growth conditions'). Cultures were then diluted 100× into C+Y medium without IPTG and grown until the $OD_{595\,nm}$ of the wt strain reached 0.2. At this point, cultures were diluted to $OD_{595\,nm}$ of 0.1 and incubated for 45 min at 30°C in order to slow down growth prior to imaging. 4 ml of cell suspension was spun down at 10,000 g, and pellets were fixed in glutaraldehyde solution 2.5% (EMS) Electron microscopy sciences and in osmium tetroxide 1% (EMS) with 1.5% of potassium ferrocyanide (Sigma) in phosphate buffer (PB 0.1 M [pH 7.4]) for 1 hr at room temperature. Samples were washed twice with $H_2O$, and pellets were embedded in agarose 2% (Sigma) in water and dehydrated in acetone solution (Sigma) at graded concentrations (30%–40 min; 70%–40 min; 100%–2 × 1 hr). This was followed by infiltration in Epon resin (EMS) at graded concentrations (Epon 33% in acetone – 2 hr; Epon 66% in acetone – 4 hr; Epon 100%–2 × 8 hr) and finally polymerized for 48 hr at 60°C. Ultrathin sections of 50 nm thick were cut using a Leica Ultracut (Leica Mikrosysteme GmbH) and picked up on a copper slot grid 2 × 1 mm (EMS) coated with a polystyrene film (Sigma). Sections were post-stained with uranyl acetate (Sigma) 4% in $H_2O$ for 10 min, rinsed several times with $H_2O$ followed by Reynolds lead citrate in $H_2O$ (Sigma) for 10 min, and rinsed several times with $H_2O$. Micrographs were taken with a transmission electron microscope FEI CM100 (FEI) at an acceleration voltage of 80 kV with a TVIPS TemCamF416 digital camera (TVIPS GmbH).

## In vivo experiments using a murine pneumonia disease model

Male C57BL/6JRj mice (8 weeks old) (Janvier Laboratories, Saint Berthevin, France) were maintained in individually ventilated cages and were handled in a vertical laminar flow cabinet (class II A2, ESCO, Hatboro, PA). All experiments complied with national, institutional, and European regulations and ethical guidelines were approved by our Institutional Animal Care and Use guidelines (D59-350009, Institut Pasteur de Lille; Protocol APAFIS#16966 201805311410769_v3) and were conducted by qualified, accredited personnel.

Mice were anesthetized by intraperitoneal injection of 1.25 mg (50 mg/kg) ketamine plus 0.25 mg (10 mg/kg) xylazine in 200 µl of PBS. Mice were infected intranasally with 30 µl of PBS containing 50 plaque-forming units of the pathogenic murine-adapted H3N2 influenza A virus strain Scotland/20/74 (*Matarazzo et al., 2019*). Seven days later, mice were inoculated intranasally with $10^5$ CFU of *S. pneumoniae* strain 19F in 30 µl of PBS. Mice were treated intragastrically with 5 mg of clomiphene (Clomid, Sanofi-Aventis, France) in 200 µl of water and or 1 mg of amoxicillin (Clamoxyl for injection, GlaxoSmithKline) in 200 µl of water at 8 and 12 hr post-infection, respectively. This amoxicillin concentration was chosen because it is expected to lead to in vivo amoxicillin concentrations above the MIC of the resistant 19F strain and because we need to see an antimicrobial effect of the antibiotic alone on the bacterial load to then be able to see any synergistic effect with another compound. Mice were sacrificed 24 hr post-infection by intraperitoneal injection of 5.47 mg of sodium pentobarbital

in 100 µl PBS (Euthasol, Virbac, France). BAL fluids were sampled after intratracheal injection of 1 ml of PBS and centrifugation at 1400 rpm for 10 min. Lungs and spleen were sampled to determine the bacterial load. Tissues were homogenized with an UltraTurrax homogenizer (IKA-Werke, Staufen, Germany), and serial dilutions were plated on blood agar plates and incubated at 37°C. Viable counts were determined 24 hr later. Statistical significance between groups was calculated by the Kruskall-Wallis test (one-way ANOVA). Analyses were performed with Prism software (version 9, GraphPad Software, La Jolla, CA).

## Acknowledgements

We thank Mark van der Linden, German National Reference Center for Streptococci for providing us with the *S pneumoniae* clinical isolates of Spain-23F and 11 A strains and Frédéric Wallet, Regional Reference Center for pneumococci, Lille, for providing the *S pneumoniae* strain 19 F. We are thankful to the electron microscopy facility of the university of Lausanne for their assistance in obtaining TEM images. We thank Emanuele Cattani for technical assistance, construction of strains VL3710, VL3711 and VL3712, and microscopy data used in Figure 3—figure supplement 1B-C. Work in the Veening lab is supported by the Swiss National Science Foundation (SNSF) (project grants 310030_200792 and 310030_192517), SNSF JPIAMR grant (40AR40_185533), SNSF NCCR 'AntiResist' (51NF40_180541) and ERC consolidator grant 771534-PneumoCaTChER. LD is the recipient of a Marie Skłodowska Curie Actions – Individual Fellowship (MSCA-IF, proposal number 837923). JCS, MB, and CC received funding from Région Hauts de France STaRS, and the European Union's Horizon 2020 research and innovation program under grant agreement No 847,786.

## Additional information

### Funding

| Funder | Grant reference number | Author |
|---|---|---|
| European Commission | Marie Skłodowska Curie 837923 | Liselot Dewachter |
| European Research Council | 771534-PneumoCaTChER | Jan-Willem Veening |
| Schweizerischer Nationalfonds zur Förderung der Wissenschaftlichen Forschung | 310030_200792 | Jan-Willem Veening |
| HORIZON EUROPE Framework Programme | 847786 | Jean-Claude Sirard |
| Schweizerischer Nationalfonds zur Förderung der Wissenschaftlichen Forschung | 310030_192517 | Jan-Willem Veening |
| Schweizerischer Nationalfonds zur Förderung der Wissenschaftlichen Forschung | 40AR40_185533 | Jan-Willem Veening |
| Schweizerischer Nationalfonds zur Förderung der Wissenschaftlichen Forschung | 51NF40_180541 | Jan-Willem Veening |

The funders had no role in study design, data collection and interpretation, or the decision to submit the work for publication.

## Author contributions
Liselot Dewachter, Conceptualization, Data curation, Formal analysis, Funding acquisition, Investigation, Methodology, Project administration, Supervision, Validation, Visualization, Writing - original draft, Writing - review and editing; Julien Dénéréaz, Formal analysis, Investigation; Xue Liu, Formal analysis, Investigation, Methodology; Vincent de Bakker, Formal analysis; Charlotte Costa, Mara Baldry, Investigation; Jean-Claude Sirard, Funding acquisition, Methodology, Project administration, Writing - review and editing; Jan-Willem Veening, Conceptualization, Funding acquisition, Project administration, Supervision, Writing - review and editing

## Author ORCIDs
Liselot Dewachter  http://orcid.org/0000-0003-1080-1656
Xue Liu  http://orcid.org/0000-0001-6485-1865
Vincent de Bakker  http://orcid.org/0000-0003-1019-3558
Jan-Willem Veening  http://orcid.org/0000-0002-3162-6634

## Ethics
All experiments complied with national, institutional and European regulations and ethical guidelines, were approved by our Institutional Animal Care and Use guidelines (D59-350009, Institut Pasteur de Lille; Protocol APAFIS#16966 201805311410769_v3) and were conducted by qualified, accredited personnel.

## Decision letter and Author response
Decision letter https://doi.org/10.7554/eLife.75607.sa1
Author response https://doi.org/10.7554/eLife.75607.sa2

---

# Additional files

## Supplementary files
- Supplementary file 1. TableS1: sCRilecs-seq results.
- Supplementary file 2. TableS2: GO enrichment of sCRilecs-seq hits.
- Supplementary file 3. Table S3: sCRilecs-seq results for mevalonate mutant.
- Supplementary file 4. Table S4: GO enrichment of sCRilecs-seq hits of mevalonate mutant.
- Transparent reporting form

## Data availability
Sequencing data is available at SRA under accession number PRJNA763896. Much of this work is based upon microscopy and snap shots and movies of most experiments are included in the manuscript and supporting files. Raw microscopy images are available at the BioImage Archive (accession number S-BIAD477).

The following datasets were generated:

| Author(s) | Year | Dataset title | Dataset URL | Database and Identifier |
|---|---|---|---|---|
| Veening J-W | 2021 | Amoxicillin-resistant Streptococcus pneumoniae can be resensitized by targeting the mevalonate pathway as identified by sCRilecs-seq | https://www.ncbi.nlm.nih.gov/bioproject/PRJNA763896 | NCBI BioProject, PRJNA763896 |
| Veening J-W, Dewachter L | 2022 | Amoxicillin-resistant Streptococcus pneumoniae can be resensitized by targeting the mevalonate pathway as indicated by sCRilecs-seq | https://www.ebi.ac.uk/biostudies/BioImages/studies/S-BIAD477 | BioStudies, S-BIAD477 |

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
