## [Editor Report]

This paper will have a high impact on screening strategies, on factors that determine cell shape and peptidoglycan synthesis in pneumococcus and other bacteria, and on antibiotic potentiation to overcome resistance.

---

## [Decision Letter]

**Decision letter after peer review:**

Thank you for submitting your article "Amoxicillin-resistant Streptococcus pneumoniae can be resensitized by targeting the mevalonate pathway as indicated by sCRilecs-seq" for consideration by eLife. Your article has been reviewed by 3 peer reviewers, including Malcolm Winkler as the Reviewing editor and Reviewer #1, and the evaluation has been overseen by Gisela Storz as the Senior Editor. The following individual involved in review of your submission has agreed to reveal their identity: Carol Gross (Reviewer #2).

Essential revisions:

Three experts in the field reviewed your manuscript from slightly different perspectives. All three reviewers are generally positive about your interesting, well-presented paper and think that it leads to several advances in the field. However, the reviewers also think changes can be made that would considerably strengthen the current version and its impact. Most of these changes can be made from the existing data set and expanded discussion of methods, phenotypes, and context as described below, without additional experiments. Several relatively simple additional experiments are suggested that would contribute to some interpretations.

The three reviews offer many specific and constructive suggestions to improve the manuscript. Some of these suggestions are described in the public reviews, and others are listed in the reviews for the authors. We request a revision that addresses each major point raised in the reviews. We also request that you consider performing some of the additional suggested experiments.

Briefly, the revisions concern four topics. First, Reviewers #1 and #2 noted that the method seems incomplete and not optimized. In particular, Reviewer #2 provides several suggestions for ways to re-analyze the data to improve hit detection and reproducibility. Second, the reviewers would like more discussion of how other hits, besides the mevalonic acid pathway, may impact pneumococcal cell length. This discussion could include any new hits turned up by changes to the data analysis. Third, the reviewers request points of clarification about the mevalonic acid depletion pathway and its effects on cell shape (e.g., width by Reviewers #1 and #3) and peptidoglycan synthesis, including larger fields of cells from the microscopic analyses. These modifications may include any addition hits turned up by re-analysis of the SCRilecs-seq data during mevalonate limitation. Last, more context and comparison are requested for the potentiation studies of clomiphene on amoxicillin killing of amoxicillin-resistant S. pneumoniae strains, in view of the previous studies in *Staphylococcus aureus*.

Finally, a couple of relatively simple additional experiments are suggested to answer several questions about conclusions (e.g., live-dead staining of mevalonate depleted cells in culture compared to time-lapse; dependence of lysis on LytA hydrolase; addition of exogenous undecaprenol; perhaps (not so simply) determination of relative Lipid-II amounts; etc.). Adding some of these additional experiments would strengthen certain conclusions and should be considered for the revision.

*Reviewer #1 (Recommendations for the authors):*

Specific comments for the authors.

1. Line 26, 88, 334, 377, and throughout. The phrasing of "insufficient transport of cell wall precursors" is a bit confusing. In this case, there is reduced availability of precursors that limits PG synthesis, but the transport process (i.e., flipping) is normal. It gets confusing, because "insufficient transport" seems to imply that something is wrong with transport per se, rather than reduced amount of precursor. Please consider revising.

2. More might be mentioned in the Discussion on how to optimize the screen more. Given the context and goals of this paper, this does not need to done here, but it would be nice to see some ideas about optimization.

3. Line 228. I think that a 1.4X increase in width is more than slight, when its overall effect on cell volume is considered. Please consider re-phrasing.

4. Line 243. Please see above. What is the evidence for many survivors? Was live/dead staining and/or microscopy done to check for lysis. Why is there so much greater lysis on pads than in liquid culture? In line 245, are suppressors even possible for this essential pathway in S. pneumoniae? I suggest you tie together more the different observations mentioned in this paragraph.

5. Line 265. Please mention in liquid culture here, since the lower panel is on pads.

6. Line 314. Was the lysis dependent on the LytA hydrolase, whose activity is induced by antibiotics and PG stress? This would be interesting to know.

7. Line 375. I am not sure that the phenotype is that profoundly different at 3 and 3.5 h. Please consider how this would affect your conclusion?

8. Line 382. Please mention that possible mechanisms for this apparent threshold effect are taken up in the Discussion. The idea of a simple catalytic model for the two modes of PG synthesis is just one of several models discussed.

9. Line 383. Please cite 41-43 here instead of just 43. References 41 and 42 show a remarkably similar elongation without constriction phenotype for gpsB deletion mutants, which is accounted for by a switching model.

10. Line 446 and elsewhere. The short-hand here is a bit misleading: "large fraction" and "small fraction" mean something other than "large-cell fraction" and "small-cell fraction." Please consider changing elsewhere as well.

11. Line 546. Since the 19F strain is resistance to amoxicillin (Table 1) in vitro, why does the bacterial load go down two orders of magnitude for amoxicillin in the model in Fig. 6I? Please clarify

12. Line 575. Is the statement about no false positives entirely correct? The number of candidates reported here is relatively small (at 17), and there were many false negatives as discussed in the text.

13. Line 636. Please consider changing "a D39V lab strain" to "an amoxicillin-sensitive, virulent serotype 2 strain". Unlike true lab strains, D39 has retained strong virulence in infection models and D39 strains that were stored separately decades ago have very similar genome sequences.

14. Line 641. Based on activity and structural modeling of the WT and mutant PBPs are there references that would support this speculation? If so, please cite them.

15. Line 18. Please consider revising "cluster nicely together".

*Reviewer #2 (Recommendations for the authors):*

To improve the impact of this paper, I suggest that the authors concentrate their efforts in three major directions. First, they must improve their sCRilecs-seq method and/or analysis to improve hit detection and reproducibility. Second, the authors should discuss other genes/pathways that contribute to changes in cell size. Finally, the authors should discuss the findings around clomiphene in the context of the previous work on *B. subtilis* and MRSA. What aspects are similar? What aspects are different? What does this tell us about Streprococcus biology?

The authors' data exhibits extreme variability that may be due to issues with how the analysis and/or FACS was performed. This variability must be addressed for this method to be of general utility.

1. High variability covers up what seem to be legitimate hits. For example, sorting Table S1 by L2FC reveals many hits that are not significant at p = 0.1 (a very generous threshold) but very likely represent real hits - glmS, murB, ddl, ftsAZ, divIC, etc. Clearly the statistical treatment is incorrect if you are not detecting any false positive hits at p< 0.1, a threshold at which you would naively expect ~10% false positive hits.

a. I'm surprised that the distribution of L2FC is centered around zero. It suggests that a WT cell has an equal chance of ending up in the middle 70% bin or in the top 10% bin, and this makes no sense. How does the WT FCS distribution compare with the library FCS distribution?

b. Consider alternate metrics beside L2FC.

c. Consider outlier detection and removal.

d. I'm not sure how you normalized counts (sum of reads?), but consider other methods of normalizing counts.

2. Replicate 1 seems to be particularly problematic in the initial screen. What happened?

3. For validation, the authors should confirm the morphology of select strains under the microscope in addition to evaluation by flow cytometry. Consider sampling not only the most significant hits, but also less significant hits to validate statistical threshold.

4. Data is not presented in a usable form. The supplementary tables include "baseMean", which I assume (I shouldn't have to assume...) is the average read count in the middle 70%, and average L2FC, which is a measure of enrichment. There is no information on the individual replicates, which precludes any reanalysis of the data or interrogation of reproducibility. These issues are exacerbated by the fact that the SRA ascension includes only fastq files. Why not included a matrix of sgRNA counts for each sample as in the Cell Host and Microbe paper (Liu et al., 2021)?

5. The analysis is, in my opinion, insufficiently well described. For example, how was the data normalized between high/middle/low? Was it by the sum of total reads? Is this a reasonable assumption?

6. The methods constantly reference data sorted based on fluorescence, but I don't see any of this data in the manuscript or elsewhere.

a. L781 "highest or lowest FSC/GFP/mCherry"

b. L763 "gated based on FSC, SSC and GFP fluorescence"

A discussion of other hits in the authors' screen would dramatically improve this manuscript, especially if the issues with false negatives can be resolved. This has the potential to be a major part of the manuscript, as cell size determination has not been comprehensively explored in Streptococcus.

1. The authors have previously determined the growth of this library in vitro and in vivo. Is there an evidence of any relationship between growth rate and cell size?

2. Induction of the stringent response causes cells to become very small. Do the authors see evidence of stringent response regulators/effectors in their data?

3. Do the authors see evidence of moonlighting metabolic enzymes/pathways that may determine cell size?

4. In most bacteria the SOS system caused filamentation. As far as I know Streptococcus lacks a canonical SOS response and uses the competence regulon for this purpose. Is there filamentation when the competence system is disturbed?

5. Are there any genes of unknown and/or hypothetical function with size phenotypes? Can these phenotypes be used to help determine function?

6. The mechanical stress of breaking up chains can reveal "fragile" strains (they disappear from the population faster than expected from previous experiments). Were there any fragile strains identified in your assay? Were fragile strains associated with a particular size/morphology?

A major finding is that clomiphene potentiates amoxicillin in Streptococcus. The novelty of this finding is of questionable given that clomiphene was shown to potentiate aminopenicillins (ampicillin) in both MRSA and *B. subtilis* (Farha et al., 2015, PNAS) and to have effects on cell size.

1. The authors should expansively discuss (Farha et al., 2015, PNAS) and place their results in the context of this work. Are the interactions they discover consistent with this manuscript, or are there Streptococcus-specific differences?

2. In vitro data showed that only high concentration of clomiphene (8 μg/ml) can

sensitize S. pneumoniae to amoxicillin. Is it because of the low efficacy of clom?

3. The authors did a second screen to study the genetic interaction between the

mevalonate pathway and other division and cell wall related genes, and identified targets like DivIV and Pbp2x in Figure 5J. But it is still unclear what the "underlying genetic network" is. If the authors would like to claim a buffering or synergistic genetic

interaction, they should quantify and compare between the fitness/phenotype of the

individual and double mutants.

4. It is still not clear why disruption of the mevalonate pathway affects cell division more than cell elongation. The authors document that PG synthesis is reduced, and suggest that division is more sensitive than elongation to this reduction. This is potentially an important contribution of the ms. Are there additional experiments that could buttress it? Do any of the "small" colonies in the 2nd screen shed light on the nature of this requirement?

---

## [Author Response]

Reviewer #1 (Recommendations for the authors):Specific comments for the authors.1. Line 26, 88, 334, 377, and throughout. The phrasing of "insufficient transport of cell wall precursors" is a bit confusing. In this case, there is reduced availability of precursors that limits PG synthesis, but the transport process (i.e., flipping) is normal. It gets confusing, because "insufficient transport" seems to imply that something is wrong with transport per se, rather than reduced amount of precursor. Please consider revising.

To remove all confusion regarding this statement, we have rephrased it at all locations where it previously occurred.

2. More might be mentioned in the Discussion on how to optimize the screen more. Given the context and goals of this paper, this does not need to done here, but it would be nice to see some ideas about optimization.

As suggested by reviewer 2, we have invested time and effort into improving our data analysis pipeline which we believe was highly successful and already resulted in a large increase in significant hits (from 17 to 70 hits). Additionally, we have added some suggestions for further optimization in our discussion section at lines 677 and 692. Briefly, besides using an unencapsulated mutant as suggested by the reviewer, we believe better screening results could be obtained by choosing phenotypes to assess wisely. In contrast to FSC values that display a large spread and do not show an on-off response, other phenotypes that are better separated and display less spread should decrease the experimental variation which would lead to more accurate hit calling. Additionally, phenotypes that are not connected to essential genes would be better suited to study through sCRilecs-seq since they are unaffected by the low read counts (and corresponding weak statistical power) associated with essential genes.

3. Line 228. I think that a 1.4X increase in width is more than slight, when its overall effect on cell volume is considered. Please consider re-phrasing.

We have removed any adjectives indicating that this increase in width is small from the paragraph starting at line 281. Indeed, an increase by a factor 1.4x cannot be considered small. However, we had originally chosen this phrasing because the change in cell length is larger still. We believe that the adjustments to the text now make it clear that both changes are considerable, but that cell length is disproportionately affected.

4. Line 243. Please see above. What is the evidence for many survivors? Was live/dead staining and/or microscopy done to check for lysis. Why is there so much greater lysis on pads than in liquid culture? In line 245, are suppressors even possible for this essential pathway in S. pneumoniae? I suggest you tie together more the different observations mentioned in this paragraph.

When growing depletion strains of operon 1 or operon 2 (native operon deleted and replaced by an IPTG-inducible copy elsewhere in the genome) in the absence of the inducer we indeed see a decrease in OD that levels out, like shown in Figure 3E. However, when following growth for a period of more than 10 hours, we see a rapid increase in OD after this time window (Figure 3 – Figure Supplement 2A, see below). A successive round of growth of the resulting culture in the absence of IPTG no longer shows the expected drop in OD (Figure 3 – Figure Supplement 2C-F, see below), indicating that suppressor mutations arose that cause the culture to no longer be dependent upon IPTG for growth.

When performing time lapse microscopy with these depletion strains, we have never observed suppressor mutants that were able to grow and thrive in the absence of IPTG. However, time lapse microscopy interrogated the behavior of only a small number of cells, while the inoculum size for the OD measurements is estimated to be around 10^4^ cells, meaning that it is much more likely to encounter suppressor mutants there.

When a deletion strain of operon 1 (native operon deleted, no inducible copy elsewhere in the genome) is grown in the absence of mevalonic acid, cultures are driven to full extinction as is shown in Figure 3I. Here, the expected decrease in OD is not followed by a phase of growth, also not when the incubation period is extended (Figure 3 – Figure Supplement 2B, see below).

Because suppressor mutations are readily obtained in the depletion strains (with inducible complementation construct) but not in the deletion strain (absence of the operon in question), we believe that suppressors allow for constitutive expression of the inducible P*_lac_* promoter (e.g. mutations in *lacI* or P*_lac_*). Indeed, sequencing of a couple of suppressors of depletion of mevalonate operon 1, operon 2 or *uppS* indeed revealed mutations in *lacI* or the *lac* operator of the P*_lac_* promoter (Figure 3 – Figure Supplement 2G, see below).

Given these experimental results, we are convinced that the plateau in OD that occurs in the depletion strains of operon 1 and 2 is due to the presence and growth of suppressor mutants that are unaffected by the absence of IPTG. Since these suppressors cannot arise in the deletion strain, no plateau is found here and instead lysis occurs until full extinction. Because of these results, we strongly suspect that when performing a Live/Dead staining on the depletion versus deletion strains, the former will consist of a mix of dying cells and healthy suppressors, while the latter will only contain dying cells. In other words, we expect that the same differences between the depletion and deletion strain that are seen in OD curves will be perpetuated in a Live/Dead staining due to the presence and absence of healthy suppressor, respectively. This is why we do not believe that performing a Live/Dead staining will be an added value in this particular setting.

The results containing suppressor mutations have been added to our revised manuscript as supplemental information.

5. Line 265. Please mention in liquid culture here, since the lower panel is on pads.

“Liquid culture” has been included in the description for this figure panel (line 323).

6. Line 314. Was the lysis dependent on the LytA hydrolase, whose activity is induced by antibiotics and PG stress? This would be interesting to know.

We have investigated this question and now show that lysis is partially, but not completely dependent on LytA. These data are shown below and are included in our manuscript in Figure 3 – Figure Supplement 1F-G and mentioned in the main text at line 293.

7. Line 375. I am not sure that the phenotype is that profoundly different at 3 and 3.5 h. Please consider how this would affect your conclusion?

Of course we respect the reviewer’s point of view, however, we do believe that these two phenotypes (depletion of *mraY*, related to PG synthesis, versus depletion of SPV_1620, related to TA synthesis) differ fundamentally, with the latter leading to more variation in morphology than is seen when *mraY* is depleted. We hope that the reviewer is more convinced of this difference when seeing the larger fields of view shown below. To also make these images available to the reader, we have deposited all of them on the EMBL-EBI BioImages Archive: https://www.ebi.ac.uk/biostudies/studies/S-BIAD477.

**Author response image 1. sa2fig1:** *mraY* (top) and SPV_1620 (bottom) depletion phenotypes at 3h depletion compared to each other. Scale, 10 µm.

In case the reviewer is not convinced of the difference between both strains, this would impact our conclusions in the following way. Right now, we interpret the phenotypic data to mean that cell elongation caused by depletion of the mevalonate pathway is caused by a decrease in the availability of peptidoglycan precursors only. This also implies that peptidoglycan precursor levels are critical in regulating cell elongation versus cell division. If, however, also depletion of SPV_1620 would be interpreted to lead to the same elongation phenotype, we would conclude that decreases in both peptidoglycan precursors and teichoic acid building blocks would lead to cell elongation and a specific block in cell division, thereby implying that also teichoic acid precursors serve a regulatory role. However, as said, we are not convinced that this is the case and would therefore prefer to keep our manuscript unaltered on this point.

8. Line 382. Please mention that possible mechanisms for this apparent threshold effect are taken up in the Discussion. The idea of a simple catalytic model for the two modes of PG synthesis is just one of several models discussed.

We have now referred to our discussion for further elaboration on the possible explanations underlying this threshold effect (line 451).

9. Line 383. Please cite 41-43 here instead of just 43. References 41 and 42 show a remarkably similar elongation without constriction phenotype for gpsB deletion mutants, which is accounted for by a switching model.

We have added these references at the requested position.

10. Line 446 and elsewhere. The short-hand here is a bit misleading: "large fraction" and "small fraction" mean something other than "large-cell fraction" and "small-cell fraction." Please consider changing elsewhere as well.

We appreciate that this can indeed be very confusing and have rephrased these statements here (line 524) and elsewhere in the text.

11. Line 546. Since the 19F strain is resistance to amoxicillin (Table 1) in vitro, why does the bacterial load go down two orders of magnitude for amoxicillin in the model in Fig. 6I? Please clarify

We see a decrease in the bacterial load because we achieve in vivo amoxicillin concentrations above the MIC of the resistant 19F strain with the dose that we used in this experiment, which is 1 mg of amoxicillin. In fact, with this treatment regimen, we expect the serum concentration to reach antibiotic concentrations 10x higher than the MIC for this strain which is approximately 1.5 µg/ml and to be just above the MIC in the lungs. We commonly use this dose of 1 mg of amoxicillin in our lab for resistant models, because we need to see an effect of the antibiotic alone on the bacterial load to then be able to see any synergistic effect with another compound. This is now clarified in the Methods of the revised manuscript starting at line 1063.

12. Line 575. Is the statement about no false positives entirely correct? The number of candidates reported here is relatively small (at 17), and there were many false negatives as discussed in the text.

In our previous analysis, we validated all 17 significant hits and found that all of them increased FSC values, the characteristic on which they were selected. We are therefore confident that in our original analysis, no false positives were detected.

However, our improved analysis now results in 70 hits that are significantly enriched in the fraction of the population with the largest cell sizes. This number of hits is too high for all of them to be individually confirmed, so we are no longer certain that no false positives are detected. We have therefore removed statements regarding the absence of false positives from our manuscript.

13. Line 636. Please consider changing "a D39V lab strain" to "an amoxicillin-sensitive, virulent serotype 2 strain". Unlike true lab strains, D39 has retained strong virulence in infection models and D39 strains that were stored separately decades ago have very similar genome sequences.

This change has been applied at line 787.

14. Line 641. Based on activity and structural modeling of the WT and mutant PBPs are there references that would support this speculation? If so, please cite them.

Indeed, there are studies that indicate that mutant PBPs have altered activities that lead to fitness costs and alterations in the peptidoglycan structure. We have now slightly elaborated the discussion on this topic and have included several relevant references, starting at line 798: “It has been shown that many resistance-conferring PBP mutations are associated with considerable fitness costs that need to be compensated for and that mosaic PBPs significantly alter the cell wall structure, potentially by a decrease in affinity for their natural substrate. We therefore speculate that these mosaic PBPs display suboptimal activity which allows them to remain active during amoxicillin treatment but fail to carry out their task if the amount of peptidoglycan precursors available to them becomes limiting due to a deficiency in Und-P production.”

15. Line 18. Please consider revising "cluster nicely together".

We have changed the phrasing of this sentence in Supplementary information line 18.

Reviewer #2 (Recommendations for the authors):To improve the impact of this paper, I suggest that the authors concentrate their efforts in three major directions. First, they must improve their sCRilecs-seq method and/or analysis to improve hit detection and reproducibility. Second, the authors should discuss other genes/pathways that contribute to changes in cell size. Finally, the authors should discuss the findings around clomiphene in the context of the previous work on *B. subtilis* and MRSA. What aspects are similar? What aspects are different? What does this tell us about Streprococcus biology?The authors' data exhibits extreme variability that may be due to issues with how the analysis and/or FACS was performed. This variability must be addressed for this method to be of general utility.1. High variability covers up what seem to be legitimate hits. For example, sorting Table S1 by L2FC reveals many hits that are not significant at p = 0.1 (a very generous threshold) but very likely represent real hits - glmS, murB, ddl, ftsAZ, divIC, etc. Clearly the statistical treatment is incorrect if you are not detecting any false positive hits at p< 0.1, a threshold at which you would naively expect ~10% false positive hits.

We agree that the high variability in our screen likely covers up a lot of legitimate hits. We have therefore optimized our analysis according to the reviewer’s suggestions as also detailed in the responses below.

However, we do believe that a considerable portion of the detected variability stems from the biological data itself and is not merely caused by factors that are within our control such as data acquisition and analysis. We have chosen to sort cells based on cell size, which is not a fixed property and varies already by a factor 2 for wild-type cells during their natural cell cycle. Moreover, for a sgRNA to be detected as leading to an increase in cell size, it should not only cause lots of cells to have sizes that are within the top 10% FSC bin, but it should simultaneously deplete the number of cells that end up in the control population (middle 70% bin). Genes that upon depletion lead to a large variation in cell size might therefore automatically be missed.

Additionally, cell size is not directly measured and instead is evaluated by a cell’s forward scatter (FSC), which is a rough and imperfect measure for cell size. As is also clear from our answer to comment 1a below, the FSC distribution of the CRISPRi library remains largely unchanged by induction of the CRISPRi system, indicating that the induced CRISPRi library does not have a considerable larger spread of FSC values than a normal wild-type population would have. This relatively small impact of genome-wide gene depletions on FSC values in the populations surely complicates accurate sorting of the desired phenotype and subsequent hit calling.

Another difficulty in extracting all relevant genes involved in cell size regulation is that many of them are core essential genes. CRISPRi strains that repress such essential genes are depleted in any condition. So, in those cases specifically, we are contrasting low counts with low counts. As long as the strain count is not actually zero, we can still compute the fold change, but because of the low sequencing depth for such strains, the certainty with which we can estimate that fold change decreases, resulting in higher p-values. Hence, it is troublesome to reliably detect all genes involved in cell size regulation because many of them are essential for growth and are very quickly depleted from the entire population.

Nonetheless, we have optimized our data analysis pipeline to improve the results presented in our manuscript. To do so, we have re-extracted the sgRNA counts from the fastq files with 2FAST2Q, a new method recently developed in our lab (https://www.biorxiv.org/content/10.1101/2021.12.17.473121v1). This method is now also part of our standard CRISPRi-seq methods pipeline, which has recently been published (https://doi.org/10.1038/s41596-021-00639-6). This resulted in slightly higher coverage, while preserving only high-quality reads (PHRED score > 30). Because higher coverage means more statistical power, we expected to increase the probability of detecting real hits. Indeed, redoing the enrichment analysis with these newly acquired counts boosted the number of hits, including likely biologically meaningful ones as the reviewer suggested.

Additionally, we usually only consider sgRNAs to be hits when the log2FC exceeds a certain threshold. However, we realized that with only 3.5 hours of induction and quite large natural variability in cell size, we are also interested in relatively mild enrichment effects. Therefore, we have lowered the log2FC threshold to test against any increase/decrease in CRISPRi strain abundance (instead of a minimum of 2-fold enrichment/depletion), which allows us to detect more subtle effects. Simultaneously, we have lowered alpha to more conventional certainty (alpha = 0.05), meaning that hits are significant if the adjusted p-value < 0.05.

Taken together, these adjustments to our analysis yielded a total of 70 significant hits for which gene repression leads to an increase in cell size. These hits include all 17 sgRNAs previously identified. Moreover, *glmS*, *ddl*, *divIC* which were suggested to be important by the reviewer are now also detected as hits, as are many other. The other suggestions, *murB* and *ftsAZ*, are not detected because counts are extremely low for these operons (see remark on low sgRNA counts above).

The results generated by our new and improved analysis are included in our revised manuscript in both the results and discussion session at several locations in the manuscript.

a. I'm surprised that the distribution of L2FC is centered around zero. It suggests that a WT cell has an equal chance of ending up in the middle 70% bin or in the top 10% bin, and this makes no sense. How does the WT FCS distribution compare with the library FCS distribution?

The vast majority of sgRNAs indeed have a log2FC of zero, meaning that they are found in the same proportions in the middle 70% and the top 10% bin. This might be surprising when operating under the assumption that the top 10% bin would only contain mutant cells with very drastic increases in cell size/FSC that are unimaginable for a wild-type cell to obtain.

However, this is not the case. Wild-type cells can very easily be found in the top 10% bin as is also clear from panel A shown in the figure below. In this panel, the FSC distribution for the CRISPRi library with (green) or without (yellow) induction of the CRISPRi system is shown. Since these two curves are almost indistinguishable, it is safe to say that even without external disturbances to morphology-related genes, cells can obtain phenotypes that render FSC values found in the top 10% bin.

We believe that this can – at least in part – be explained by the fact that FSC is only a very rough measure of cell size. Indeed, when looking at changes in FSC before and after separation of cell chains (Figure 2 – Figure Supplement 1C), FSC is only modestly affected while the changes in the length of chains are considerable (Figure 2 – Figure Supplement 1A-B).

We therefore conclude that the imperfect link between cell size and FSC, combined with the naturally occurring variation in cell size/FSC for wild-type cells can explain why the log2FoldChange for many sgRNAs is zero.

b. Consider alternate metrics beside L2FC.

We want to quantify differences in strain abundance between two conditions. The log2FC captures this quantity exactly, while also neatly making the distribution symmetric around zero, and is therefore our preferred metric. In addition, this kind of quantification, and corresponding hypothesis testing techniques, have been improved to near perfection by dedicated research groups. One of those analysis packages is DESeq2, which is one of the most commonly used packages worldwide. It takes into account the nature of the data (discrete counts) by implementing a GLM based on a negative binomial distribution and has robust internal normalization methods. For these reasons, it is the standard downstream analysis method in our recently published CRISPRi-seq protocol which we would prefer to implement here.

c. Consider outlier detection and removal.

Outlier detection and removal is automatically performed by DESeq2 based on the Cook’s distance metric (measuring change in GLM coefficient estimates upon removal of each sample). In the new analysis, with 2FAST2Q-generated sgRNA counts, no such outliers were detected. As stated before, this does not mean all sgRNAs with high variability are considered outliers and they are thus not necessarily excluded from downstream analysis. Instead, their p-values simply inflate, since with higher variability among replicates, we can be less certain of any measured difference of fold change.

d. I'm not sure how you normalized counts (sum of reads?), but consider other methods of normalizing counts.

The DESeq2 normalization method is based on size factors. These are normalization factors that account for both library size bias and library composition bias, while division by the sum of reads only accounts for the former. The latter refers to the issues that arise when some sgRNAs use up either a lot or very few reads in a sample or in all samples belonging to one growth condition. This then leaves either very few or a lot of reads to be mapped to the other sgRNAs, thus yielding a bias in the whole library composition on top of the differences in just plain library size. This has been a well-documented source of bias, and most standard (RNAseq) analysis packages account for it (e.g. DESeq2, limma, edgeR). Hence, DESeq2’s size factors do not only correct for the number of reads per sample, but also for the variability in sgRNA counts over all samples, using the geometric mean. This makes the method more robust and better at detecting true differences, since the sgRNAs become more fairly comparable over conditions. For details, we refer to the original DESeq2 research paper and documentation (http://bioconductor.org/packages/release/bioc/vignettes/DESeq2/inst/doc/DESeq2.html). For these reasons, we prefer to keep using the built-in DESeq2 normalization method.

For the sake of completeness, another normalization approach was used for the PCA. There, we use DESeq2’s regularized logarithm (rlog) normalization, which is recommended and part of the standard DESeq2 pipeline. It is akin to a log2 transformation, but decouples the variance from the mean; a well-known but generally undesired statistical property yielding large spread for lower counts on a logarithmic scale, largely caused by random noise instead of biological processes. The transformation is based on a shrinking approach presented in the DESeq2 research paper, which proportionally shrinks lower counts, thereby stabilizing the variance, i.e. rendering the data homoscedastic. This is ideal for input into downstream analyses, such as PCA.

2. Replicate 1 seems to be particularly problematic in the initial screen. What happened?

We are unaware of any differences between the different repeats. Care was taken to ensure sampling and cell sorting proceeded as similar as possible and all processing steps after sorting (library preparation, sequencing, etc.) was done for all repeats in parallel. We can therefore not give any reasonable explanation for why certain repeats behaved differently.

3. For validation, the authors should confirm the morphology of select strains under the microscope in addition to evaluation by flow cytometry. Consider sampling not only the most significant hits, but also less significant hits to validate statistical threshold.

As suggested, we validated hits using microscopy. Because of our improved analysis and increased number of hits, we only tested sgRNAs that were found to be among the most strongly enriched significant hits. Results are shown below and are included in our manuscript in Figure 2. Images that correspond to this quantitative analysis of cell size have been deposited at the EMBL-EBI BioImages Archive: https://www.ebi.ac.uk/biostudies/studies/SBIAD477.

4. Data is not presented in a usable form. The supplementary tables include "baseMean", which I assume (I shouldn't have to assume...) is the average read count in the middle 70%, and average L2FC, which is a measure of enrichment. There is no information on the individual replicates, which precludes any reanalysis of the data or interrogation of reproducibility. These issues are exacerbated by the fact that the SRA ascension includes only fastq files. Why not included a matrix of sgRNA counts for each sample as in the Cell Host and Microbe paper (Liu et al., 2021)?

The baseMean is the average of the normalized counts taken over all samples and is part of the standard DESeq2 output. We acknowledge that this was not clear in our original manuscript. Since it is not relevant to the interpretation of the results, we left it out in the updated differential enrichment analysis output table. In addition, we added the raw read counts per sample (2FAST2Q output) in this table and thank the reviewer for the useful suggestion.

5. The analysis is, in my opinion, insufficiently well described. For example, how was the data normalized between high/middle/low? Was it by the sum of total reads? Is this a reasonable assumption?

We originally did not include an explanation of the analysis pipeline since this is part of the DESeq2 R package that is explained in the corresponding research paper. However, we appreciate that a rough overview of the different steps in the DESeq2 analysis is important for the understanding of our work. We have therefore included a brief explanation on the inner workings of DESeq2, including the normalization method, in our Materials & Methods section, starting at line 940: “Briefly, DESeq2 takes raw count data as input and fits a negative binomial generalized linear model (GLM) to them. It normalizes by size factors, which account for both library size and library composition bias. Next, it tests for differential enrichment of each sgRNA between given conditions by testing the hypothesis that the log2 fold-change (the fitted sgRNA-wise GLM coefficients) significantly differs from a user-specified threshold. Here, we tested for any differential enrichment (absolute log2FC > 0) at a significance level of 0.05 (alpha = 0.05). Thus generated sgRNA-wise p-values were adjusted for testing multiple comparisons with the Benjamini & Hochberg correction, controlling the false discovery rate (FDR). We added the sample information to the design formula of DESeq2 to account for differences between samples (paired data, as the Min, Middle and Max fractions were taken from the same samples).”

6. The methods constantly reference data sorted based on fluorescence, but I don't see any of this data in the manuscript or elsewhere.a. L781 "highest or lowest FSC/GFP/mCherry"b. L763 "gated based on FSC, SSC and GFP fluorescence"

At line 950 comment a, the text has been changed to now only list FSC as a parameter used for sorting. Including GFP and mCherry was a mistake and a remnant from an old version of this manuscript where we also included screening data where cells were sorted based on DNA content (HlpA-GFP) and Z-rings (FtsZ-mCherry).

At line 924 comment b, the original text was correct. Cell gates were set based on FSC, SSC and GFP with the latter stemming from the HlpA-GFP fusion. Whereas it may not have been completely necessary to have a gate based on GFP fluorescence and therefore DNA content, we believe it did lower noise since it enabled us to more clearly separate cells from dust particles and other impurities in the cell suspension.

A discussion of other hits in the authors' screen would dramatically improve this manuscript, especially if the issues with false negatives can be resolved. This has the potential to be a major part of the manuscript, as cell size determination has not been comprehensively explored in Streptococcus.

We have now included such a discussion in our revised manuscript. We discuss the types of hits that are found in our screen and highlight a few interesting findings as well as listing hypothetical proteins that we now implicate in cell size regulation. This text can be found in our results section starting at line 221 and is also mentioned in our discussion.

1. The authors have previously determined the growth of this library in vitro and in vivo. Is there an evidence of any relationship between growth rate and cell size?

Unfortunately, strains and conditions for this screen deviate from previous in vitro screens that we performed (https://doi.org/10.1016/j.chom.2020.10.001). First, in our previous CRISPRi-seq screens (where the CRISPRi-seq library was grown in the presence or absence of the inducer of *dcas9* expression and then sequenced to determine the relative sgRNA abundances), the library was grown for approximately 9 hours before sequencing. However, this long growth step results in the (almost) complete depletion of many sgRNAs that target essential genes. As said before, these extremely low counts would lower statistical power in our current experimental set-up which is why we decided to drastically shorten the incubation period to 3.5 hours. Additionally, the strain used here for library construction encodes the HlpA-GFP and FtsZ-mCherry fusion proteins which are absent from our previously constructed library. Whereas neither HlpA-GFP nor FtsZ-mCherry cause any growth defects, it is possible that they do show synergies and/or antagonisms when combined with CRISPRi-imposed gene depletions. Because of these reasons, comparing the CRISPRi screening results regarding growth rate and cell size might be tricky and we would prefer to do this as part of future work when the exact same libraries and growth conditions are used.

2. Induction of the stringent response causes cells to become very small. Do the authors see evidence of stringent response regulators/effectors in their data?

We specifically looked at the abundances of sgRNA0557 that targets *relA*, the bifunctional (p)ppGpp synthetase/hydrolase of *S. pneumoniae.*

**Author response table 1. sa2table1:** 

sgRNA	Target	Log2FC	Adjusted P value	Screen	Contrast
sgRNA0557	relA	-1,611951038	0,188689	1^st^, wt	Max_vs_Middle

CRISPRi-dependent inhibition of *relA* expression leads to a relative depletion of these cells in the fraction of the population with the highest FSC values, indicating that cells may become smaller when *relA* is depleted. However, this difference is not statistically significant. We are therefore hesitant to draw any conclusions regarding the effect of *relA* depletion on cell size.

3. Do the authors see evidence of moonlighting metabolic enzymes/pathways that may determine cell size?

To properly answer this question, we would have to relate growth rate to cell size and as explained above we are very reluctant to do so with the data sets we currently have. However, this is a very interesting question that we will try to answer in the future.

4. In most bacteria the SOS system caused filamentation. As far as I know Streptococcus lacks a canonical SOS response and uses the competence regulon for this purpose. Is there filamentation when the competence system is disturbed?

We have looked at the effect of downregulation of several key competence genes on *S. pneumoniae* cell size as measured by our sCRilecs-seq screening approach.

**Author response table 2. sa2table2:** 

sgRNA	Target	Log2FC	Adjusted P value	Screen	Contrast
sgRNA0688	SPV_2427, comGG, comGF, comGE, comGD, comGC, comGB, comGA	-0,54399	0,817735	1^st^, wt	Max_vs_Middle
sgRNA0764	tRNA-Glu-5, comE, comD, comC1	-1,16648	0,134294	1^st^, wt	Max_vs_Middle
sgRNA0863	comX2, comX1	-0,07914	0,963779	1^st^, wt	Max_vs_Middle
sgRNA0868	SPV_1623, srf-03, comA, comB	0,74547	0,677056	1^st^, wt	Max_vs_Middle

Changes in FSC values upon depletion of parts of the competence regulon are inconsistent (we see both enrichments and depletion in the fraction of the population with the highest FSC values) and are in general associated with very high p values. This is also the case for our second screen using a Δmev strain (results shown in Supplementary File 3). We therefore conclude that downregulation of competence genes most likely does not influence cell size. However, it remains possible that induction of the competence regulon would affect size and morphology. We do not expect competence to be induced under our experimental conditions as we used acid C+Y (Slager et al. 2014 Cell). This is a question we cannot answer with our current data set. However, it is well documented that competence induction leads to cell elongation (Berge et al. 2017 Nature Comm).

5. Are there any genes of unknown and/or hypothetical function with size phenotypes? Can these phenotypes be used to help determine function?

We were very happy to see that our screen is able to pick up several hypothetical proteins.

**Author response table 3. sa2table3:** 

sgRNA	Target	Log2FC	Adjusted P value	Screen	Contrast
sgRNA1362	SPV_1931	4,237775065	0,025273781	1^st^, wt	Max_vs_Middle
sgRNA0046	SPV_0131	3,45244365	0,000257296	1^st^, wt	Max_vs_Middle
sgRNA0872	SPV_0010	2,89001648	1,99841E-06	1^st^, wt	Max_vs_Middle
sgRNA0943	SPV_0418	2,646042226	0,01539123	1^st^, wt	Max_vs_Middle
sgRNA0606	SPV_1594, SPV_1595	1,544551591	0,041585927	1^st^, wt	Max_vs_Middle

We indeed strongly believe that our results can help pinpoint the function of these genes of unknown function. As our GO enrichment analysis showed, hits are likely to be involved in processes such as cell division and cell envelope synthesis. A future first step will therefore be to study how depletion or deletion of these genes affect growth, division and cell envelope integrity. We have now mentioned these exciting new leads in the revised manuscript.

6. The mechanical stress of breaking up chains can reveal "fragile" strains (they disappear from the population faster than expected from previous experiments). Were there any fragile strains identified in your assay? Were fragile strains associated with a particular size/morphology?

Since we cannot directly compare our screening results to previously obtained data where no rigorous vortexing was performed, we cannot give a comprehensive answer to this question. However, when performing validation of selected hits by microscopy, cells were vortexed before visualizing them. This did not lead to clear traces of lysis, meaning that vortexing is unlikely to have strong effects on the outcome of the screening. Images of the microscopy validation have been deposited on the EMBL-EBI BioImages Archive: https://www.ebi.ac.uk/biostudies/studies/S-BIAD477.

A major finding is that clomiphene potentiates amoxicillin in Streptococcus. The novelty of this finding is of questionable given that clomiphene was shown to potentiate aminopenicillins (ampicillin) in both MRSA and *B. subtilis* (Farha et al., 2015, PNAS) and to have effects on cell size.1. The authors should expansively discuss (Farha et al., 2015, PNAS) and place their results in the context of this work. Are the interactions they discover consistent with this manuscript, or are there Streptococcus-specific differences?

We believe we have successfully addressed this comments by discussing and comparing both the antimicrobial and potentiation effects of clomiphene (paragraph starting at line 778) and the effect of clomiphene on *B. subtilis* morphology (starting at line 705).

2. In vitro data showed that only high concentration of clomiphene (8 μg/ml) cansensitize S. pneumoniae to amoxicillin. Is it because of the low efficacy of clom?

Indeed, a clomiphene concentration of 8 µg/ml is most potent at boosting the antimicrobial activity of amoxicillin in vitro. However, we do see some effect also at lower clomiphene concentrations. At a concentration of 4 µg/ml, for example, the sensitivity to amoxicillin of all tested strains has increased by a factor 2. For clinical isolates that are resistant to amoxicillin some very minor effects can even be seen at the lowest tested clomiphene concentration of 0.5 µg/ml.

However, to obtain an effect that might be of clinical importance we indeed need a concentration of 8 µg/ml which is quite high. We therefore hope to improve the compound in the future to increase its efficacy and lower the concentration needed to potentiate amoxicillin.

3. The authors did a second screen to study the genetic interaction between themevalonate pathway and other division and cell wall related genes, and identified targets like DivIV and Pbp2x in Figure 5J. But it is still unclear what the "underlying genetic network" is. If the authors would like to claim a buffering or synergistic geneticinteraction, they should quantify and compare between the fitness/phenotype of theindividual and double mutants.

We have studied and presented the effect of a select few double mutants in our manuscript. We have also investigated a couple more. However, having these double depletions of essential genes makes experiments very tricky and, for most double depletions, we were unable to identify combinations of inducer concentrations that led to wild-type behavior. This absence of proper control conditions prevented us from conducting reliable experiments which is why we decided to not include many double depletion mutants in our manuscript. However, we do believe that in our revised manuscript we have answered the question on the mechanism underlying the mevalonate elongation phenotype better than we previously did. More specifically, we interpreted our results – including from our second screen – in light of the new model for peptidoglycan synthesis in ovococci recently published by the Morlot group (https://www.sciencedirect.com/science/article/pii/S0960982221005765). We believe this represents a significant improvement to our first manuscript version.

4. It is still not clear why disruption of the mevalonate pathway affects cell division more than cell elongation. The authors document that PG synthesis is reduced, and suggest that division is more sensitive than elongation to this reduction. This is potentially an important contribution of the ms. Are there additional experiments that could buttress it? Do any of the "small" colonies in the 2nd screen shed light on the nature of this requirement?

We agree that this is a very important point for follow-up studies. We believe these should be focused on determining whether or not the rate of peptidoglycan splitting influences the mevalonate elongation phenotype. Our work presents the first support for this hypothesis since we have identified the proposed peptidoglycan splitting enzyme PcsB as a hit in our screen. Our results indicate that depletion of *pcsB*, and likely corresponding slowing of septal cell wall splitting, prevents excessive cell elongation upon mevalonate depletion. This is also highlighted in our manuscript. Further work is needed to dive into these observations and investigate these observations in more detail.